



# Quantifying economic risks to dairy farms from volcanic hazards in Taranaki, New Zealand

Nicola J. McDonald[1], Leslie Dowling[1], Emily P. Harvey[1], Alana M. Weir[2,3], Mark S. Bebbington[4], Nam Bui[1], Christina Magill[5], Heather M. Craig[2], Garry W. McDonald[1], Juan Monge[1], Shane J. Cronin[6], Thomas M. Wilson[2], and Duncan Walker[7]

[1]ME Research, Takapuna, Auckland, New Zealand.
[2]University of Canterbury, Christchurch, New Zealand
[3]University of Geneva, Geneve, Switzerland
[4]Massey University, Palmerston North, New Zealand
[5]GNS Science, Wellington, New Zealand
[6]University of Auckland, Auckland, New Zealand
[7]Perrin Ag Consultants Ltd, Rotorua, New Zealand

**Correspondence:** N. McDonald (nicky@me.co.nz)

**Abstract.** The volcanic hazard and risk science for Taranaki Mounga (Taranaki volcano) in New Zealand is in an advanced state, with robust probabilistic data and a series of direct impact scenarios modelled for the region. Here we progress this work and demonstrate a method to provide risk information that is nuanced for factors such as location and economic sector, and considers the dynamic nature of volcanism with hazards potentially repeated over time. Recognising the fundamental

importance of the dairy sector to Taranaki Region, this paper provides valuable insights on the potential impacts and risks to heterogeneous dairy cattle farms within the region from volcanic hazards. We provide volcanic impact and risk metrics in economic or monetary terms in order to improve its relevance to decision makers while reducing the complexity of the impacts. To do this, we developed a dynamic, multi-event farm system model of response and recovery, which takes in hazard intensity metrics from a series of volcanic events, and generates the resulting annualised revenues, expenditures and recovery

costs through time. The model is formulated in a generalised way such that it can be used for various other hazard types and agricultural land uses. In our application of the model, we create and apply a suite of ten thousand simulations that capture different iterations of possible future volcanic activity over a 50-year period. These include the generation of lahars following eruptions and associated failures for transport and water supply networks. Farms at five case study locations were modelled, to capture the diversity in farm management and the spatial variation in hazard intensities and likelihoods across the region. We

provide summaries of the distributions of economic impacts generated, both for individual events and for the 50-year volcanic future horizon. Drawing the information together, we also summarise the results for each case study farm in terms of the Value at Risk statistic. For the case study farms with negligible lahar risk we find, with 90% confidence, that volcanic losses over the next 50 years will not exceed around 10% of property value. By comparison, for the farm with the most severe lahar and ashfall exposure, we find that at the same level of confidence, losses extend to approximately half the property value. These

results indicate that with access to sufficient risk information, we should anticipate volcanic risk as playing an important role in shaping the future dairy sector in Taranaki Region. The modelling pipeline and assessment metrics demonstrated in this paper





could be used to assess mitigation and adaptation strategies to reduce the risk from volcanic hazards and improve the resilience of farm businesses.

# 1 Introduction

Taranaki Mounga is a stratovolcano located in the western North Island of New Zealand. The probability of a volcanic event occurring has been estimated at between 33% and 42% over the next 50 years (Damaschke et al., 2018). Thus, for the surrounding region, there is a distinct possibility of a volcanic event impacting economic activities, including the more than 1,000 dairy farms in the vicinity of the mountain.

Although it is generally recognised that there is a smaller body of disaster risk science for volcanic hazards than other
hazards, especially when compared to the more frequently occurring weather related hazards (Ward et al., 2020), the hazard and risk assessment for Taranaki has a comparatively high level of research that has gone into understanding geological, engineering and social aspects of risk. A body of research has helped to quantify eruption frequencies and magnitudes along with the variety of physical processes and consequential hazard types (see Cronin et al. (2021)), while several studies have built knowledge of the vulnerabilities in socio-ecological systems (e.g. Wilson et al. (2009); Wild et al. (2019); Weir et al.
(2024b)) and started to quantify the impacts of potential eruption scenarios (Weir et al., 2022, 2024a; McDonald et al., 2017). Nevertheless, providing information on the dynamic impacts and risks to socio-economic activities that is nuanced for factors such sector, location, timing and management approach, and in a form that is appropriate to and meaningful for decision makers, is an ongoing topic of research.

In this study we focus on potential volcanic impacts and risks to dairy cattle farms within Taranaki Region, recognising that
dairy cattle farming is the dominant land use surrounding the volcano and the dairy sector is strongly embedded in the regional economy directly constituting approximately 13% of gross regional product (Cardwell et al., 2023). While it is feasible that over the near to medium economic future (say 20-50 years) no volcanic events will occur, it is also possible that there will be several events. We know that the avenues by which dairy farms may be impacted from volcanic events are various and complex, including multiple types of hazards (e.g. ashfall, lahars-volcanic mudflows and floods) and hazards that manifest
at the physical location of the farms as well as impacting on networks and infrastructure supporting farms, such as transport and water supply networks (Deligne et al., 2022). The effects from potentially multiple eruptive events or hazards may occur separately, or overlap (Lawrence et al., 2020). Some impacts will be immediate and short term, while others will persist over time. Furthermore, heterogeneity across dairy farms (Doole and Pannell, 2012) and in human responses and adaptation will potentially influence the way in which impacts manifest.

Providing volcanic impact metrics in economic or monetary terms is one tactic to help reduce complexity and assist decision-makers in making sense of potential impact and risk information. Monetary metrics can aggregate across direct and indirect, immediate and future, and short- and long-term impacts. In this study, we propagate the impacts of volcanic events through individual farm system business models of response and recovery. The model evaluates economic impacts considering both farm enterprise asset stocks and changes in expenditure and revenue flows. The use of such system-based models of enterprises





is advantageous in terms of familiarity to stakeholders, having been used to quantify economic impacts for numerous policy and
risk assessment contexts in New Zealand, including the assessment of options to reduce agriculture greenhouse gas emissions
and improve freshwater quality (see, for example, Parsons et al. (2015), Journeaux et al. (2020) and He Waka Eke Noa (2022)).
Because the model provides a systemic view of farm operations, it is also possible to connect it to wider economic models that
trace flow-on impacts through an economy (Monge et al., 2023; Hallegatte, 2008; Monge and McDonald, 2020; Brown et al.,
2019a).

In summary, the purpose of this paper is to demonstrate how we can provide valuable insights on the potential economic
impacts and risks to land based enterprises around volcanic regions. To achieve this, we develop a dynamic business impact
and recovery model that has broad applicability to volcanic as well as other hazard types, and demonstrate its application in
a modelling pipeline that quantifies economic impacts across many simulations, and ultimately derives risk-based metrics.
The example we have chosen as a case study is dairy cattle farms within Taranaki Region in New Zealand. The following
sections of this paper begin by providing a general description of the dynamic business impact model developed for calculating
economic impacts of hazard events at the farm scale (Section 2), and further details of the Taranaki case study (Section 3).
This is followed (Sections 4 and 5) by explanation of the way in which the model was set up for the case study in terms of
scenarios considered and data and parameter setting. In the results and discussion section (Section 6), the economic impacts
of individual volcanic events and over 50-year simulations are presented along with summaries of risk in terms of probability
density functions and the Value at Risk statistic.

## 2   Agricultural Business Behaviours Model

The Agricultural Business Behaviours Model (AgriBBM) operates at a single farm level, inputting hazard intensity metrics (or
metrics describing intensity of cascading failures such as water supply and transport disruption) from a series of events to work
out the resulting annualised revenue, expenditure, and costs (REC) through time. In describing the model, we use notation in
a way that we intend to be generalisable to multiple hazard types and agricultural land uses. The details developed for the
specific case of dairy farms under volcanic hazards, including some necessary adjustments and extensions for the specific case
study region, are covered in sections 4.3-4.6, and Appendices A to C.

In this section, we will first step through the model for a single event, then describe the development of the agricultural
business recovery trajectory function, and finally describe how we deal with multiple, potentially overlapping, events in a
single scenario. An oveview of the model is shown in Figure 1.

### 2.1   AgriBBM steps for a single event

Each farm is modelled independently. For each farm, we start with a vector of base farm accounts $F_{i,j}$, where $j$ is the index for
each line item of the annual revenue and expenditure for business as usual, and $i$ is the year. The annual Farm Operating Profit
is calculated from $\sum_j F_{i,j}$. In general, these accounts can be set to vary by year $i$. This allows for the inclusion of policies that
change fees or subsidies through time, or for farms where the revenue and expenditure depends on the age of the land use. This



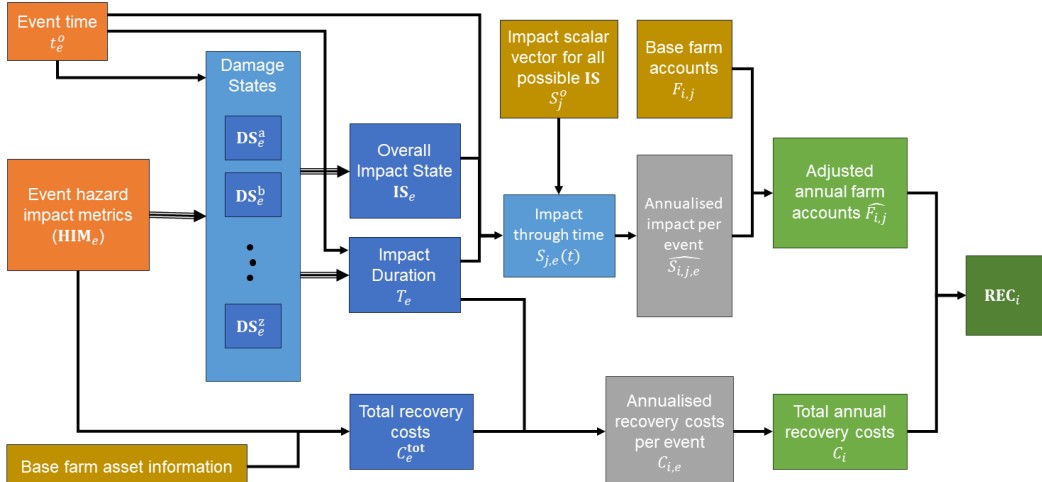

**Figure 1.** Agricultural Business Behaviours Model overview diagram for a single farm, showing the progression from the hazard event information, $\mathbf{HIM}_e$, and time, $t_e$ for events $e$ (orange), through to the annual Revenue, Expenditure, and Costs, $\mathbf{REC}_i$ (green). For each farm the model needs inputs (brown): base accounts at line item level for each year $F_{i,j}$, base farm asset information, e.g. building replacement cost, and an estimate of the relative impact scalars for each overall Combined Impact State (**IS**) at line item level, $S_j^0$.

is relevant for agricultural industries such as horticulture where it can take a number of years for plants to get established, or forestry, which have very different annual expenditure and revenue depending on age of the plantation.

As shown in Figure 1, we need to consider the disruptions to the farm's operation and behaviour, and its recovery trajectory
due to an event at time $t^0$, as well as estimating the recovery costs associated with the event.

First, we use the hazard intensity metrics to determine the Damage States (**DS**) for the event split up by each type of damage or disruption. The term "Damage State" is chosen here simply for convenience. Note that we do allow for the individual categories and associated functions that could be considered here to include those relating to damages to capital as well as those describing the loss or disruption in the provision of a service, e.g. $\mathbf{DS}^{\text{ashfall}}$ for the Damage State due to direct ashfall on
the farm impacting on pastures, crops, stock, buildings and other assets and $\mathbf{DS}^{\text{water}}$ for the Damage State due to outages in the water supply to farms. Such Damage States are typically determined by probabilistic fragility functions and the set of Damage States considered will depend on the type of hazard being modelled, as well as the farm types being considered.

The next step is to combine the Damage States into an overall Combined Impact State (**IS**) for that event. By allowing for multiple Damage States, and then combining them into a single overall Combined Impact State, this framework can be
generalised to different hazards and different agricultural land uses. For each overall Combined Impact State **IS** we need to consider how a farm would respond. For most farm types, an event will also impact their ability to operate as usual, and they will have to adjust to a different mode of operation. In many cases this will result in a combination of increases in expenses, e.g. having to buy in extra feed, as well as a decrease in income, e.g. due to loss of harvest or inability to get products to market when subject to a transport disruption. These changes are expressed as a vector of scalars, $S_j^0$ (where $j$ is the index of the line





item in the farm accounts), that specifies the maximum relative change in each revenue/expenditure line item from BAU ($F_j$) due to this behaviour change. This will be a key part of developing the model for a specific farm type and case study region.

At this step, we also need to estimate the Impact Duration $T$, which will depend on the overall Impact State as well as potentially the magnitude of the different hazard impacts. Note here that the Impact Duration is not the duration of the acute hazard event, or even how long the farm's behaviour is affected, it is the duration of time that the annualised farm accounts

are affected. As an example, if there was a disruption that was only 1 week long, but it meant that a horticultural farm lost its whole harvest, the impact on the annualised accounts would need to be for a whole year, i.e $T = 1$ year.

For our model, we need to calculate the annualised impact on the accounts. For a given year this means working out the proportion of the year impacted by the event, and the mean value of the impact scalar $S_j(t)$ during that time.

The mean value of the impact scalar, during the period of each year that the event impacts is:

$$\widehat{S}_{i,j} = \frac{1}{\Delta t_i} \int_{a_i}^{b_i} S(t)\,dt \tag{1}$$

where $a_i$ is that start of the event impact in year $i$ ($a_i = \min(t^0, i-1)$), $b_i$ is the end of the event impact in year $i$ ($b_i = \max(t^0 + T, i)$), and the duration of the impact in year $i$ is $\Delta t_i = b_i - a_i$. Whether the resulting integral can be evaluated explicitly depends on the form of the recovery function.

The resulting annual farm operating accounts, after adjustments due to the hazard impact, for year $i$ of the simulation are

given by:

$$\widehat{F}_{i,j} = F_j \times \left[ \left( \widehat{S}_{i,j} \Delta t_i \right) + (1 - \Delta t_i) \right] \tag{2}$$

where $j$ is the index for the farm accounts line items. Note that for all non-impacted periods, $S_j = 1$.

When a farm is impacted by a natural hazard event, in addition to the disruption and changes to business operation, there will often be recovery costs. These can include clean up costs as well as replacement of damaged assets. We need to determine

the total recovery costs $C^{\text{tot}}$ and for this estimate, we will need farm specific information, such as building replacement values, effective hectares, and stock numbers. We then spread the total recovery cost $C^{\text{tot}}$ out evenly over the impact duration. $T$, to produce an annualised recovery cost, $C_i$, for year $i$:

$$C_i = \begin{cases} C^{\text{tot}}, & \text{if } i-1 < t^0 \leq i \quad \text{and} \quad t^0 + T \leq i \\ \frac{C^{\text{tot}}(i - t^0)}{T}, & \text{if } i-1 < t^0 \leq i \quad \text{and} \quad t^0 + T \geq i \\ \frac{C^{\text{tot}}(i - (t^0 + T))}{T}, & \text{if } t^0 \leq i-1 \quad \text{and} \quad t^0 + T \leq i \\ \frac{C^{\text{tot}}}{T} & \text{otherwise} \end{cases} \tag{3}$$

The output of the model is a timeseries of the annual Revenue, Expenses, and Costs at line item level $\mathbf{REC}_{i,j}$, for each farm

at each location, which is created by appending the recovery costs $C_i$ onto the Farm Operating Accounts $\widehat{F}_{i,j}$. The Annual Net Revenue, $\mathbf{REC}_i$, is then $\sum_j \mathbf{REC}_{i,j}$.



To consider impacts that may be far into the future, the standard economic practice is to 'discount' future cashflows by reducing future cashflows according to a rate of discount over time. This leads to the Discounted Revenue, Expenses, and Costs:

$$\mathbf{DREC}_{i,j} = \frac{\mathbf{REC}_{i,j}}{(1+r)^i} \tag{4}$$

where $r$ is the annual discount rate, and $i$ is the year in the future, with $i = 0$ being the point in time for which we are calculating present values. Finally we calculate the Net Present Value (**NPV**):

$$\mathbf{NPV} = \sum_{i,j} \mathbf{DREC}_{i,j}$$

## 2.2 Business Recovery

Consistent with a preceding Business Behaviours Model (Brown et al., 2019b), which in that case focused on urban business recovery following an earthquake, businesses are conceived as following a pathway of recovery that is defined by a recovery curve or function across time. However, unlike the Brown et al. (2019b) function, which is applied only to the modification of business outputs or revenues, we require a function that can be equally applied to expenditures. Importantly, depending on the expenditure type, expenditures may potentially be both higher and lower than business-as-usual during the recovery phase. Furthermore, given the range of potential magnitudes for volcanic and other events, it is imperative that the function is flexible to allow for minor as well as severe impacts, and short to long recovery durations. Added to this, we prefer a function that is straightforward to evaluate on an annual basis, and it must work in the model for events that happen partway during a year and/or events that have recovery durations completed partway during a year.

Given these considerations, we have developed the impact scalar recovery function:

$$S_j(t) = S_j^0 + (1 - S_j^0)exp\left(\frac{k}{T}\Big[t - (t^0 + T)\Big]\right) \quad \text{for} \quad t^0 < t < t^0 + T \tag{5}$$

where $t^0$ is the start of the event in continuous time e.g. 4.82, $T$ is the impact duration, $k$ is the steepness of the recovery, and $S_j^0$ is the initial maximum value of the relative scalar for farm account line item $j$. Figure 2 demonstrates the shape of this recovery function for a range of values of $T$, $k$, and $S_j^0$, and includes a comparison with the existing urban Business Behaviours Model (from Brown et al. (2019b) and including modifications described in (Smith et al., 2019)).

Whether a recovery curve should follow a concave *down* decreasing or concave *up* decreasing path has been debated in the literature and will likely depend on the characteristics of the event and sectors impacted (Jonkeren and Giannopoulos, 2014). Given the nature of our case study, with agriculture businesses often facing degraded natural and other capital from which recovery cannot be initiated quickly, and with the inability of agriculture businesses to action recovery through relocation, we select a function that captures limited recovery during the initial phases of the impact duration. Nevertheless, the form of the function has some flexibility, through recovery steepness parameter, $k_e$. By setting $k_e < 10$ it can represent impacts with more immediate recoveries (almost linear)[1] and it can represent recoveries that are closer to step functions (shifting back to BAU

---

[1] For $k_e \lessapprox 5$ we start to run into an issue where $S_j(t = t_e) \neq S_j^0$, especially for shorter impact durations $T_e$, as $e^{-k_e} \approx 0$ only for $k_e \gtrapprox 5$



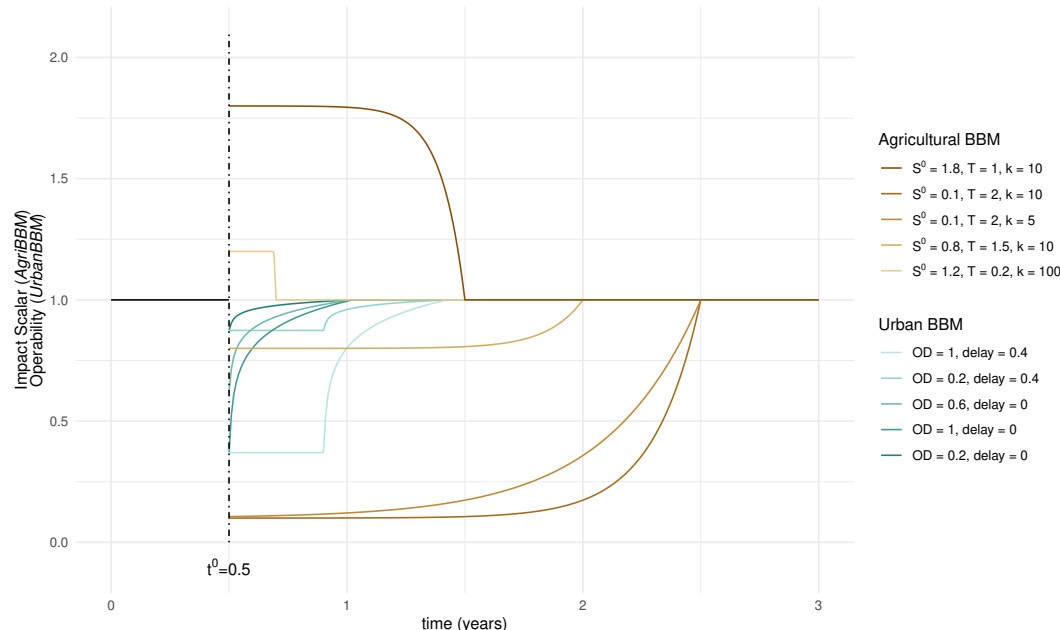

**Figure 2.** Comparison of different recovery functions for an event at $t^0 = 0.5$. In green are the operability functions from the Urban Business Behaviours Model (Urban BBM) (Brown et al., 2019b; Smith et al., 2019) for a range of 'Overall Disruption' (OD) and 'delay' values. In brown are the impact scalar curves for the new agricultural business behaviours model (AgriBBM) for a range of values for the impact scalar $S_j^0$, impact duration $T$, and recovery steepness $k$.

only at the end of the impact duration) by setting $k_e >> 10$. Of course, alternative functional forms could be investigated for other applications.

For this form of the recovery function, we are able to explicitly integrate $S_j(t)$ to calculate the average adjustment to line item $j$, during the annual accounts period for year $i$, and thus the adjusted farm accounts. Combining equations (1) and (2), we get:

$$\widehat{F}_{i,j} = F_j \times \left[ \int_{a_i}^{b_i} S_j(t)\, dt + (1 - \Delta t_i) \right]$$

For the chosen recovery function, the integral can be evaluated and is:

$$\int_{a_i}^{b_i} S_j(t)\, dt = \left[ S_j^0 t + (1 - S_j^0) \frac{T}{C} e^{\frac{C}{T}(t - (t^0 + T))} \right]_{t=a_i}^{b_i} \tag{6}$$

**2.3    Extending the AgriBBM for multiple events**

In realistic volcanic scenarios, there will often be more than one event, and the impact of events can be overlapping, e.g. the next event can occur before a farm has recovered from the first event. In our case study, for each hazard scenario, we first





split the scenario into specific discrete events $e$, that occur at times $t_e^0$, and have an associated set of hazard intensity metrics. We now need to extend the single event model to account for these multiple events. In most cases the equations are exactly

the same, only an extra subscript, $e$, added for the event number. However, when events are overlapping, the situation is more complex.

Firstly, for calculating the contributing Damage States, $\mathbf{DS}_e$, the overall Impact State, $\mathbf{IS}_e$, and the impact duration $T_e$, we consider each event independently.

For recovery costs, we make the assumption that the cost to repair or replace assets and to clean up, will be the same

regardless of any preceeding events. This means that the total annual recovery costs are $C_i = \sum_e C_{i,e}$, where equation (3) becomes:

$$
C_{i,e} = \begin{cases} C_e^{\text{tot}}, & \text{if } i-1 < t_e^0 \leq i \quad \text{and} \quad t_e^0 + T_e \leq i \\ \frac{C_e^{\text{tot}}(i-t_e^0)}{T_e}, & \text{if } i-1 < t_e^0 \leq i \quad \text{and} \quad t_e^0 + T_e \geq i \\ \frac{C_e^{\text{tot}}(i-(t_e^0+T_e))}{T_e}, & \text{if } t_e^0 \leq i-1 \quad \text{and} \quad t_e^0 + T_e \leq i \\ \frac{C_e^{\text{tot}}}{T_e} & \text{otherwise} \end{cases}
\tag{7}
$$

There is a limitation in our approach for multi-events in that it may overestimate recovery costs, particularly recovery costs that might in reality occur at a discrete points in time towards the end of recovery. Take, for example, a farm milking shed damaged

during an initial volcanic event. It is possible that a second volcanic event occurs prior to the replacement of this shed being initiated, and thus the replacement cost is only faced for the second event, whereas the model will include replacement costs for the first event. This is a topic for future model refinement, particularly if used for applications where overlapping recovery phases are prevalent in the scenarios considered.

Because the impact scalar vector $S_{j,e}^0$ for each Impact State $\mathbf{IS}_e$ represents a specific shift in behaviour for the farm, the

values for specific line items $j$ cannot be modified independently of each other. This means that combining the impact of overlapping events cannot be done line item by line item e.g. we cannot add the impacts $S_j(t) \neq S_{j,e}(t) + Sj, e+1(t)$ or take the maximum relative impact of the two event line items. To illustrate, we can consider the case where there is one event ($e = 1$) causing a farm to lose grazing area and necessitating more bought in feed to maintain same stock numbers, here $S_j^0 > 1$ for $j$="Feed costs". This can be compared to another event ($e = 2$) where the impact requires the farm to destock drastically,

reducing revenue, but also meaning the farm does not need to bring in much feed, if any, so $S_j^0 \ll 1$ for $j$="Feed costs".

For our chosen recovery function, the impact scalar at any point in time is the sum of $S_{j,e}(t) = \sum_e S_{j,e}(t)$

$$
S_{j,e}(t) = S_{j,e}^0 + (1 - S_{j,e}^0)exp\left[\frac{k_e}{T_e}\left(t - (t_e^0 + T_e)\right)\right] \quad \text{for} \quad t_e^{\text{start}} < t < t_e^{\text{end}}
$$

where $t_e^{\text{start}}$ and $t_e^{\text{end}}$ are set such that only one $S_{j,e}(t) \neq 0$ at one time, because a farm can only be in one impacted state (described by a single vector of impact state scalars) at a time.

When events overlap, we use the impact duration as a proxy for the severity of the event to determine whether the farm stays in the current state or switches to the state of the new event. This gives us $t_e^{\text{start}} = \max(t_e^0, t_{e-y}^0 + T_{e-y})\{y : T_{e-y} > T_e\}$ i.e. if





any of the preceding event(s) have a larger impact and the farm is still in recovery from it, this larger event determines the functional state of the farm until its recovery has finished. And $t_e^{\text{end}} = \min(t_e^0 + T_e, \, t_{e+y}^0)\{y : T_{e+y} > T_e\}$ i.e. when a larger event occurs during recovery, this new larger event superseded this event (see Figure 3).

Given the adjustments for when each event $e$ applies' ($t_e^{\text{start}}$ to $t_e^{\text{end}}$), we then use that period to work out $a_{i,e} = \min(t_e^{\text{start}}, i - 1))$, $b_{i,e} = \max(t_e^{\text{end}}, i)$ for each year $i$ and the mean value of the impact scalar during that period.

$$\widehat{S}_{i,j,e} = \frac{1}{\Delta t_{i,e}} \int\limits_{a_i}^{b_i} S(t)dt \tag{8}$$

where $\Delta t_{i,e} = b_{i,e} - a_{i,e}$.

Finally we calculate the adjusted annual operating accounts, taking into account all events, using:

$$\widehat{F}_{i,j} = F_j \times \left[ \sum_e \left( \widehat{S}_{i,j,e} \Delta t_{i,e} \right) + \left( 1 - \sum_e \Delta t_{i,e} \right) \right] \tag{9}$$

As before, we then create $\mathbf{REC}_{i,j}$ by appending the total recovery costs $C_i$ onto the Farm Operating Accounts $\widehat{F}_{i,j}$ and then can calculate $\mathbf{DREC}_{i,j}$ following equation (4).



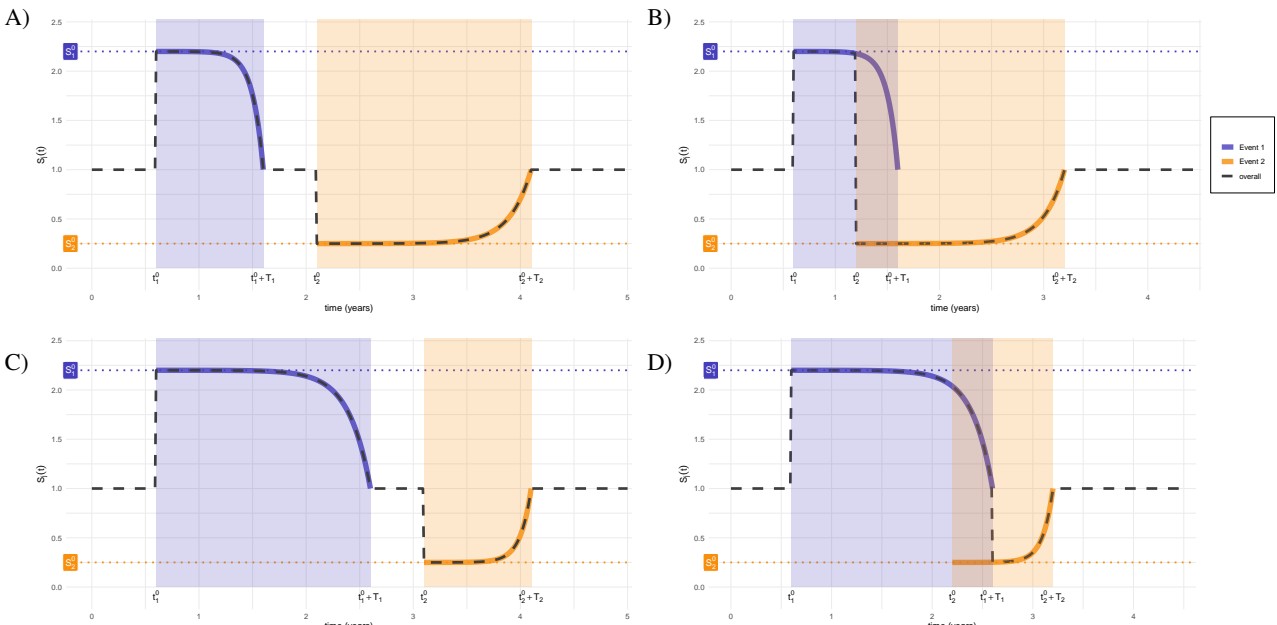

**Figure 3.** Example plots of how the impact scalar for a specific line item $S_j(t)$ changes through time when two events occur. The dashed black line shows the resulting impact scalar $S_j(t)$. In purple is the impact of event 1, $S_{j,1}(t)$, with maximum impact scalar $S_1^0$, and an impact duration $T_1$ indicated by the purple shaded period. In orange is the impact of event 2, $S_{j,1}(t)$, with maximum impact scalar $S_2^0$, and an impact duration $T_2$ indicated by the green shaded period. In panels A and B, event 2 has more impact* than event 1, so if they overlap (as in panel B), the farm state changes to the event 2 impact immediately. In panels C and D, event 1 has more impact* than event 2, and if they overlap, the farm state stays in event 1 impact until the end of the impact duration, before switching to the event 2 impact.

*Note: here we use recovery time as a proxy for impact magnitude.



## 3 Study Case

Much of the Taranaki Region is comprised of an extensive ring plain that surrounds Taranaki Mounga (Taranaki volcano) of
overlapping volcaniclastic deposit fans (Cronin et al., 2021). A National Park, Te Papakura o Taranaki, encircles the volcano
out to a radius of approximately 10 km measured from the volcanic summit. Beyond this distance the region is dominated by
agriculture, with dairy cattle farming being a major land use in the region since around the 1880s (Figure 4).

The most recent eruptive activity at Taranaki, dated as 1780-1800 AD (Lerner et al., 2019), occurred prior to the development
of large-scale agriculture in the region. Volcanism at Taranaki Mounga is characterised by cycles of edifice growth and collapse,
with growth periods containing effusive and lava dome-forming activity interspersed with explosive eruptions (Turner et al.,
2011; Lerner et al., 2019; Cronin et al., 2021). Ashfall, and the mobilisation of volcanic material in the form of lahars, are the
two hazards that contribute the most to widespread volcanic risks for agriculture within the region and thus are a focus of this
study (see Figure 4- D and Figure 6). Within the volcano's eruptive history, the hazards of lava flows and ballistic projectiles
have occurred relatively infrequently (Turner et al., 2011) and been largely contained within the vicinity of Te Papakura o
National Park, and are thus not considered in this study. While several large-scale debris avalanches have occurred, caused by
the catastrophic collapse of unstable portions of the edifice (Zernack et al., 2012), these events are very infrequent and so are
also not considered. We do not explicitly model the risks from pyroclastic density currents (PDCs). The exposure to PDCs is
less than that of ashfall and lahar hazards, but PDCs may extend beyond the park boundaries. On the other hand, the impacts
of PDCs on farming operations are expected to be similar to lahars, and the footprint of the latter is larger. Thus while initial
damage may be due to PDC , a subsequent lahar would traverse the same area and generate largely the same scale of damage
in this case

In terms of cascading failures following a volcanic event, disruptions to water supply and transport networks have been
identified as particularly important for dairy cattle farms (Wild et al., 2019). The transport network is dominated by one major
highway to each side of the mountain, only three principal access points out of the region, and numerous bridges crossing
rivers channels that have formed in a radial manner around the mountain. Since river channels are likely locations for the
flow of lahars, we have focused on nine principal channels in this study (Fig 6). The transport and water supply networks are
also subject to risk of damage and disruption from ashfall (Wilson et al., 2014). Farms within the region source water either
directly from nearby streams, from groundwater, or from community/municipal supply schemes (Wild et al., 2019). The extent
to which we expect supplies of water to be disrupted vary according to mode of supply, with those relying on surface water
abstraction most at risk of disruption (Porter, 2022).

For this study we have selected five sites at which dairy cattle farming is located for which to model the economic conse-
quences of volcanic hazards. The sites were selected to capture a diversity of risk contexts, as well as variation in the land
use capability (Lynn et al., 2009). In terms of the latter, we nevertheless note that land use capability or production capacity
does not vary significantly among the region's dairy farms, and management choices are likely to be the largest determinant of
business-as-usual variation between farm systems.



**Figure 4.** Overview of the study area: a) World map showing the location of North Island; b) North Island of New Zealand, highlighting the Taranaki Region; c) Volcanic hazards in the Taranaki Region (adapted from Neall and Alloway (1996)) with case study farm locations; d) Detailed map of the Taranaki Region, showing key urban areas (New Plymouth, Stratford, and Hawera), highways, and selected rivers for the study; ; and e) Land use in Taranaki Region.



## 4   Data and Model Set Up

### 4.1   Representative farms

We have not sought to obtain financial data for the real-life dairy farms existing at our selected study locations, as these data would in any case be subject to confidentiality. Instead the approach has been to assign base financial accounts for 'representative' farms at each location. Reference was made to a set of farm financial and environmental data collected for 49 anonymous farms in Taranaki for the 2020-21 financial year.[2] With assistance from an agricultural economist involved in the data collection, the accounts for one among the 49 farms was assigned to each study location on the basis that the selected accounts described a farm that could feasibly exist at each site. Adjustments were then made to the accounts so that revenue line items reflected long-run average commodity prices, rather than the specific prices experienced during the year for which the data was collected.[3]

The five case study farms are representative of dairy farms in the region, ranging from low to high-input dairy systems, or from system two to five as defined in Hedley et al. (2006). A summary of the key attributes of the case study farms is given in Table 1. There is a large range in cow performance in terms of milk solids produced per live weight of cows. Farm earnings also vary per hectare, even among farms of the same system type. Analogous results have been found in other studies (see, for example, Moran et al. (2019)), highlighting the importance of the unique management and operational choices in determining farm performance.

**Table 1.** Key attributes and performance indicators for the Five Case Study Farms

| Farm ID | Study location | Effective area (ha) | Dairy system type | Feed imported | Peak cows milked | Milk production (kg MS/kg LW) | Earnings (EBITDA/ha) | Water supply |
|---------|----------------|---------------------|-------------------|---------------|------------------|-------------------------------|----------------------|--------------|
| Farm 1 | A | 123 | III | 15% | 357 | 0.89 | $3,610 | Municipal |
| Farm 2 | B | 107 | IV | 25% | 244 | 0.94 | $2,920 | Municipal & Surface |
| Farm 3 | C | 73 | III | 15% | 246 | 0.86 | $3,710 | Groundwater |
| Farm 4 | D | 153 | III | 15% | 420 | 0.76 | $4,080 | Surface & Groundwater |
| Farm 5 | E | 79 | II | 9% | 205 | 0.64 | $2,590 | Groundwater |

[2]The New Zealand Ministry of Primary Industries' Farm Monitoring and Benchmarking Project - see https://www.mpi.govt.nz/funding-rural-support/farming-funds-and-programmes/farm-monitoring-and-benchmarking-project/

[3]The line item categories are consistent with those in FARMAX, a modelling tool frequently used in the industry to test the outcomes of management options aroound improving profitability and environmental performance - see https://farmax.co.nz




The five case study locations were allocated water supply types from their locations relative to municipal supply schemes and suitable surface water collection points using the approach from Wild (2016). Properties situated at a distance from municipal supply schemes and suitable surfacewater collection streams were assigned groundwater collection.

## 4.2 Volcanic Scenarios


A suite of ten thousand simulations (scenarios) of future volcanic activity, each over a 50-year period, were generated for Taranaki. Only eruptions of significant volume to extend beyond the park boundary were considered. For each simulation, the first onset time and subsequent repose times, and hence the number of events, were generated from the temporal model described by (Damaschke et al., 2018), while the magnitude of each event in terms of Volcanic Explosivity Index (VEI) (Newhall and Self, 1982) is based on the Expert Elicitation model described in Bebbington et al. (2018). The eruption volumes for each event were then obtained by probabilistic interpolation of the VEI, assuming unit differences in VEI to be equivalent to a 10-fold increase in volume.


Of the ten thousand simulations, 57% contained no volcanic event occurrence, reflecting the underlying probability of a Taranaki volcanic event over the next 50 years (Figure 5). The majority of the simulations in which volcanic activity occurs contain only one event (30% of all simulations), however, 10% contain two events and 2% three events. The greatest number of volcanic events occurring over a single 50-year simulation is six.


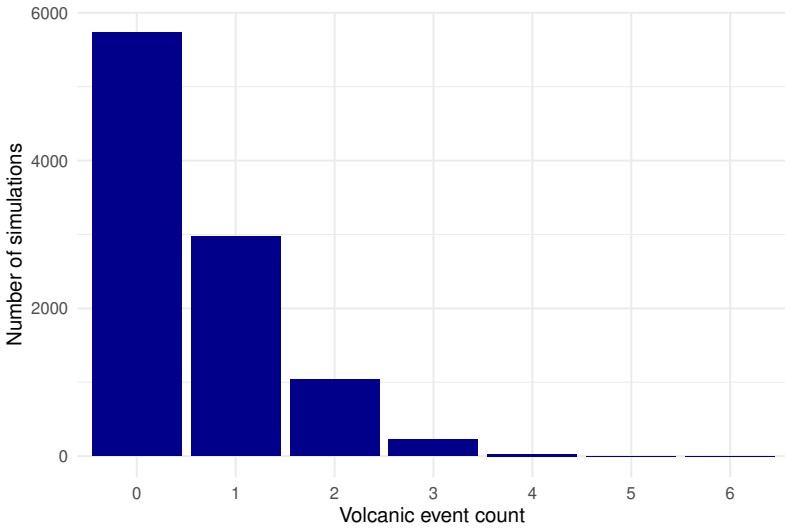

**Figure 5.** Summary of 10,000 volcanic future simulations for Taranaki showing the count of simulations by number of events.

To produce estimates of the ashfall implications of each event, we had available a pre-processed set of 500 potential Taranaki volcanic events, each with an associated spatial ashfall deposition layer for the surrounding landscape (1 km grid). These were produced via runs of the tephra2 (version 2),[4] a tephra dispersion simulation tool with wind velocities sampled from NOAA

---

[4]see https://github.com/geoscience-community-codes/tephra2





REANALYSIS database. Each event in our 10,000 simulations was matched to one of these pre-processed ashfall scenarios by sampling from within the ashfall scenarios that had an equivalent volume. To demonstrate that across all simulations ashfall is more likely to be deposited on the eastern side of the mountain due to the prevailing wind directions, Figure 6 depicts the summation of all ash deposited over all events and simulations for points on or near roads or buildings.

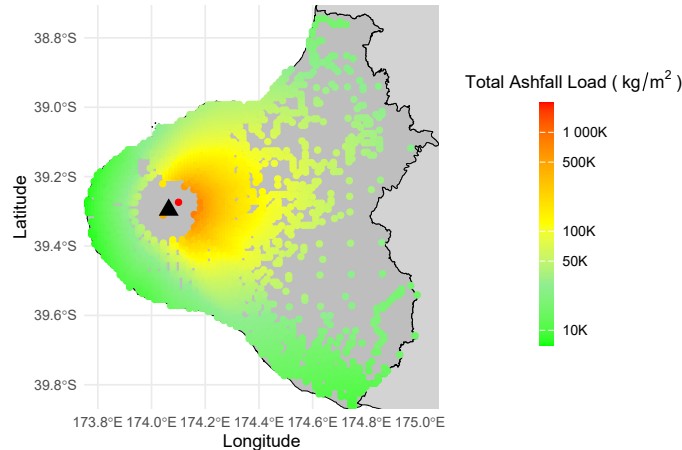

**Figure 6.** Cumulative ashfall load in the Taranaki Region across all 10,000 simulations, on a logarithmic scale.

Section 4.4 below provides further information on the processes by which the ashfall information for each event was later
converted to metrics of hazard intensity relevant to the damage and fragility functions utilised in the modelling.

### 4.3 Functions for damage states and impact states

Reflecting the principal risks to farm operations and capital, the Combined Impact State for dairy cattle farms was calculated by combining the Damage States calculated from four separate vulnerability models, each relying on inputs of hazard intensity:[5] **DS**$^\text{ashfall}$ is the Damage State for farms caused by ashfall impacting directly on pastures, crops, farm buildings and other farm
assets; **DS**$^\text{lahar}$ is the Damage State for farms caused by a lahar impacting directly on pastures, crops, farm buildings and other assets; **DS**$^\text{transport}$ is the Damage State for farms caused by experience of transportation disruption; and **DS**$^\text{water}$ is the Damage State for farms caused by experience of loss of water supply services. The derivation of the Damage States for each event was as follows:

---

[5]In the study of *natural* hazards, the term "hazard" is often reserved for the processes and phenomena principally of biophysical origin, whether occurring immediately (e.g ashfall) or cascading from an initial hazard (e.g. flooding following the silting up of river channels from volcanic material). However, if we take the broad definition of hazard such as provided by the UNDRR (United Nations Office for Disaster Risk Reduction, 2016) "a process, phenomenon or human activity that may case loss of life, injury or other health impacts, property damage, social and economic disruption or environmental degradation", cascading infrastructure failures, which might have impacts distant from the locations of the initial natural hazard, could also be considered hazards in themselves with associated 'hazard intensity metrics'. Ultimately it is a matter of preferred terminology.





- **DS**ashfall: Here we used fragility functions for pastoral farms from Craig et al. (2021). These fragility functions calculate the probabilities that large pastoral-farming business (which include dairy farms) will experience each of five damage states (refer to Appendix A1) ranging from no disruption (DS0) through to total loss of capabilities (DS4). Thus, given ashfall depth, a damage state could be assigned to each farm and event through probabilistic sampling.[6]

- **DS**lahar: Only two damage states were recognised for a farm with respect to lahars, that is DS0 (no lahar experienced) and DS4 (majority of farm directly impacted by lahar), refer to Appendix A3.

- **DS**transport: Fragility functions were constructed following the same structure as the ashfall fragility functions (see Figure A1b in Appendix A), with the independent variable being the number of days of inaccessibility experienced by the farm (ie rather than ashfall thickness). Informed by a previous ex ante study of a major hazard event that had involved identifying key metrics of transport disruption (Smith et al., 2019), the intensity metric of inaccessibility is defined as the inability to travel to any of the three points on a state highway leading into/ out of the Taranaki Region (see Figure 4). Altogether, five damage states were recognised (see Appendix A2). While the lower damage state (DS1) describes the situation of transport disruption increasing transportation costs, the higher damage states (DS2, DS3, DS4) also entail loss in the ability to supply output to market and cows dried off and/or culled (DS3, DS4). One aspect to note is that even if the farm location is never completely cut off during an event (inaccessibility duration = 0 days), there is still assigned a high probability for the event of having transport delays and additional costs (DS1), due to ashfall impacts on the wider network. A period over which vulnerability is lowered was also applied in the calculations, to reflect when cattle are not being milked.

  Independent of the ashfall related transport disruptions, we need to consider the transportation disruptions that might be caused by lahars, specifically the loss of bridges making roads impassable. For this case study we have applied a bespoke adjustment whereby if bridges crossing the principal river channels are required for access to one of the main regional access points, and these bridges are impacted by lahar, the probabilities of being in damage states were adjusted upwards. Full details of the calculations are provided in Appendix A2.

- **DS**water Only two damage states were recognised, DS0 (limited disruption) and DS4 (severe disruption). A simple threshold damage function was chosen where DS4 is assigned if the number of days of interrupted water supply for a farm is greater than a specified threshold, otherwise DS0 is assigned. Details for the calculation are given in Appendix A4.

The combination of damage states experienced by a farm determines the overall Combined Impact State of that farm. Altogether, thirteen unique impact states are recognised. At the extremes, IS0 (no impact) occurs when the damage state is DS0 across all four categories, while IS4 (severe impact, farm ceases operation) occurs when any one of the four categories are DS4. The other eleven impact states represent different combinations of ashfall and transport accessibility damage states. The complete mapping between damage states to impact states is provided in Appendix C.

---

[6]Although the authors allowed for the damage state probabilities to be adjusted downwards during times of the year with lower vulnerability, it was reasoned that the adjustment need not be applied in the study given the intensive nature of dairy farming in the region, with highly balanced operations year round, plus given the limited impact of seasonality on risks to dairy farming capital from ashfall.



### 4.4 Intensity metrics for hazards/ cascading failures

#### 4.4.1 Ashfall for DS$^{ashfall}$

For each farm and event, the ashfall thickness serves as an input to the fragility function determining **DS**$^{ashfall}$. This was determined by taking the ashfall load from the volcanic scenarios at the closest data point to the centre of the farm. The ashfall load (kPa) was then converted to thickness (mm) using a deposit density of 1000 kg/m$^3$, which is commonly used in volcanology studies of eruptions with similar compositions (Barker et al., 2019; Magill et al., 2015; Taddeucci et al., 2011).

#### 4.4.2 Lahar for DS$^{lahar}$

There is a very high likelihood of lahars being generated following an eruption event at Taranaki (Procter et al., 2020). The above described damage functions for lahars require a binary input for each event that describes whether a farm is located directly in the path of a lahar. This information was generated for each event by randomly sampling from probability distributions that relate the likelihood of lahar realisation to: (1) the size of an eruption in terms of material deposited, and (2) the proximity of the farm to the valleys or catchments in which the ash is principally deposited. For (1) the data taken for an event from the eruption scenarios is the maximum ash depth as measured around the circumference of Te Papakura o Taranaki National Park boundary,[7] while for (2) we have measured the radial degrees between the farm location and the point where the maximum ash depth is recorded around the park boundary. The matrices describing the probability distributions are contained in Appendix B1. Note that a unique distribution is provided for each farm, allowing for the probabilities to be set high for those farms located close to the mountain and where the farm, or a large proportion thereof, is located in a valley. Matrices are not provided for Farms 1 and 5, as it was concluded that the lahar risk for these locations is negligible. The assigned probabilities for the three remaining farm locations were based on expert judgement, informed particularly by the risk categories assigned to each location in the lahar hazard mapping (Neall and Alloway, 1996), as well as the results of the modelling of lahars from eruptions of several sizes and weather dispersal mechanisms by Weir et al. (2022).

#### 4.4.3 Inaccessibility for DS$^{transport}$

As explained above, inaccessibility is defined as the inability to travel to any of the three points on a state highway leading into/ out of the Taranaki Region. The road network was split into an existing road segment classification, with these segments ranging in length from one meter to 28 kilometres, with an average length of 760 meters. Segments were assigned the closest ashfall estimate from the spatially explicit eruption simulations. Network analysis was used to check for connectivity between each location and any of the three regional road entry and exit points assuming no traffic flow on road segments over a threshold of 10 mm of compacted ashfall. Based on recent experience with silt removal in New Zealand following cyclone Gabrielle (Hawkes Bay Regional Council, 2023), the rate of ash removal is set to 2400 tonnes per day across all roads in the region, and that was applied evenly over the road network. Connectivity was assessed on a daily time step for each location.

---

[7]As few ashfall measurement points are recorded exactly on the park boundary, a buffer of 3,000 m each side of the park boundary was used for collecting the ashfall data measurements.





The transport accessibility damage functions also require inputs for each event relating to lahars, in this case it is a binary input for each of the nine principal river valleys (see Figure 4) specifying whether a lahar occurs, flowing down at least as far as the respective State Highway crossings. Provided a lahar reaches this location, it is then assumed the bridge crossing will be damaged. An approach analogous to that used for farm locations was used to assign lahar hazard realisations to the river valleys - refer to Appendix B1 for the assigned probabilities. The probability distribution for Stony River is different from that

used for other rivers on the basis that the Stony River valley has a higher likelihood of lahars due to the geography of the valley and as confirmed in geological records (Procter et al., 2020).

### 4.4.4    Water Supply Disruption for DS$^{water}$

The chosen failure metric for water disruption is the days of lost service. Separate functions were developed for each water supply type to relate ashfall thickness to the estimated duration of water supply disruption, based on available literature. Inter-

ruption for surface and groundwater collection was based on ashfall thickness at the location of the farm. Days of interruption for municipal supply were based on ashfall thickness at the pump power source, water collection point, and the location of the farm. A maximum of 200 days of water distruption was assumed for large events based on the scenario in Porter (2022). Reference can be made to Appendix B2 for further information.

### 4.5    Impact scalars ($S_j^0$) and impact durations ($T$)

Impact scalars and impact durations are outlined jointly in this section as there are interrelationships between the parameters selected with the impact duration being the time over which impact scalars applied. For impact states that are likely of short duration, i.e. less than one year, it was considered convenient to generally assign an impact duration of one year and then apply impact state scalars that would be applicable when considering changes in revenue/expenditure for an entire year subsequent to the event commencement. All remaining impact states were assigned one of five approaches for estimating impact duration

- see Appendix C for the complete mapping. In summary, the approaches for estimating impact durations were as follows:

–    **Ash depth function (DF).** Where ashfall is the most impactful hazard realised, the depth of ash at the farm is used to estimate the recovery time by applying a piecewise linear function. The shape of the function, depicted in Appendix C, was informed by data gathered on recovery following the Mt Hudson and Mount St. Helens eruptions (Wilson et al., 2011a, b; U.S. Geological Survey, 2024; Dale and Crisafulli, 2018; del Moral and Wood, 1993).

–    **Inaccessibility period (IP).** Where inaccessibility is the dominant impact, the impact duration is either set simply as a defined period (e.g. 0.5 or 1 year) or the length of the inaccessibility experienced by the farm during the milking period.

–    **Water supply re-establishment (W).** Where a farm experiences DS4 for water supply disruption and all other damage states are DS2 or lower, a farm's impact duration is deemed to be controlled by the recovery of water supply. It is considered that by six months a farm will have established an alternative water supply option.





– **Maximum of ash and water (WDF).** Where a farm experiences DS4 for water supply and a high damage state for ashfall, the impact duration is set as the maximum of that calculated from the ash depth function and for water supply re-establishment.

    – **Maximum of lahar and ash (LDF).** For all impact states where a farm has experienced a lahar, the impact duration is set as either that calculated from the ash depth function or for lahar recovery, whichever is larger. An average recovery
and rebuild time for lahar is estimated to be 7.5 years, informed by literature indicating multi-year recovery timeframes for pasture and ecosystem recovery from such volcanic disturbances (Crittenden et al., 2003; del Moral and Grishin, 1999; Saputra et al., 2022).

The full set of impact scalars applied in the modeling are available as a supplementary file. Note that where a farm experiences IS0 the scalars are all set to 1, meaning no difference from business-as-usual. This can be compared to IS4, where all
scalars, except for local government rates payments which are deemed to continue, are set to zero implying a ceasing of farm operations.

The derivation of impact scalars for the remaining ashfall related impact states involved translating the qualitative descriptions of impact states provided by Craig et al. (2021) into best estimates of proportional changes in dairy farm income and expenditure items. In general, as the impact of ash increases, the quantity of feed able be grown on farm decreases, thus reduc-
ing farm revenues and increasing feed costs. At the same time, however, the number of stock remaining on farm also decreases with higher ashfall impacts, thus reducing expenditures necessary for stock maintenance. The same scalars were generally applied across all farms, except in relation to feed and grazing expenditures, as the proportion by which business-as-usual costs for these items change also depends on each farm's starting position regarding the proportion of feed sourced off farm.

For the impact state characterised by only a minor transport disruption (i.e DS1 for inaccessibility), the scalars for freight
and cartage costs were increased. For the impact states made up of more severe transport disruptions (i.e. DS2 and DS3 for inaccessibility), the scalars act to decrease farm revenues, reflecting disruption to production activities.

For impact states that involve a combination of ashfall damage and transport inaccessibility, many of the scalars selected are more severe than the worst case considering ashfall and inaccessibility impacts independently. This reflects that when multiple hazards are experienced, the options available to mitigate impacts reduces. For example, a farmer cannot simply purchase more
feed to replace that which cannot be produced on farm due to ashfall damage if transportation to the farm is also disrupted.

### 4.6 Recovery costs ($C_{i,e}$)

In addition to the disruption to "normal" revenues and expenditures, a farm also potentially faces costs to repair assets and rehabilitate land. For ashfall damage, we assigned these costs to each event according to damage functions that estimate, for each asset type, the proportion of damage experienced based on ashfall depth. Reference can be made to Appendix D for full
details. For farms experiencing a lahar, 100% damage to assets is assumed, with the same replacement and land rehabilitation costs applied as for ashfall.





A further and potentially significant cost faced by farms following an event is the cost to restock cattle. Implicit to the setting of the impact scalars that modify farm revenue and expenditure line items (see Section 4.5) was the assigned proportion of stock that can remain on each farm under each impact state (see Table C1 in Appendix C). When a farm experiences a damage state of DS2 or below for transport, it was assumed that half of the value of the stock that must be removed from the farm can be recaptured by stock sales. However, for the higher transport damage states, it was assumed that farms have no opportunity to recoup any value of the cattle that cannot be supported on-farm given the transport difficulties in getting stock to market.

## 5 Procedure

We ran the AgriBBM for each of the 10,000 volcanic scenarios and for the modelled dairy farms (i.e. Farms 1-5). We also tested locating each of the five modelled farms at each of the five study locations. Note it is not necessarily the case that each of the farms chosen as representative farms could be located at any any of the study locations, given that site characteristics were taken into consideration when selecting the representative farm for each location. Nevertheless, it was informative to test running the model for each farm at each location because it allowed for separating the implications of farm management (e.g. proportion of imported feed) from the implications purely related to farm location (e.g. proximity to vent). The mode of water supply available to a farm (e.g. surface water versus groundwater collection) was held fixed to a location. We also fixed the random numbers generated for fragility function sampling by event and location in order to best enable comparisons across farms placed at the same location.

To illustrate the results from each model run, we show fifty years of Annual Net Revenues for an example scenario in Figure 7. This scenario had three eruptions within the fifty-year simulation, occurring at 20, 22, and 42 years. The depth of ashfall and the realisation of lahars varied across the five locations. The figure depicts separately the Adjusted Annual Farm Operating Profit and the Total Annual Recovery Costs, with the summation being the Net Revenue each year.





**Figure 7.** Annual Net Revenue (NZ$) for five farms at five locations for an example scenario comprised of three eruption events. Icons represent the incidence and magnitude of volcanic impacts.

## 6 Results and Discussion

In this section, we summarise the results across the 10,000 ashfall simulations and discuss some of the findings and priorities
for future work. Although it is possible to draw out detailed (line-level) results for each farm, we concentrate on the damage
440 states and the overall revenues, expenditures and costs experienced by the farms under the various simulations.



## 6.1 Damage States

The impact state of farms was determined by combining the four damage states calculated for ashfall, lahar, transport and water supply. Figure 8 describes the distribution of damage states at each of the five farm locations, across the events. For simplicity, in undertaking this and the subsequent analysis in Section 6.2, we have only considered scenarios where there is a single event during the 50-year simulation.[8] Note that the assignment of the damage state was by location, and was independent of the farm.

The results for locations B and D are clearly dominated by the frequent incidence of lahar, with lahars associated to 37 and 66 percent of all events at these respective locations. Given that these sites are the closest to the vent, and located on the side of the mountain where ash deposition is more prevalent (especially at location D) it is not surprising that these locations also experienced the most severe damage states in terms of ashfall and transportation.

Although site E is on higher ground with negligible lahar risk, being located to the east of the mountain it does experience quite significant ashfall and transportation risk, with 35 percent of events producing a damage sate of DS2 or higher for ashfall and 50 percent DS2 or higher for transport. Located at some 37 km from the mountain, and also to the northwest of the mountain, location C experiences overall the least severe ashfall damage states. Despite non-zero probabilities for lahars, none of the simulations produced lahars affecting site C due to the lighter ashfall and ashfall-driven hazards on the western side of the mountain. Nevertheless, with many kilometres of road and channels to cross in either direction to gain access to or from the site, there were significant risks for transportation disruption. Additionally, for location A, the loss of water supply appeared to be a significant risk, with 42 percent of events returning a damage state of DS4. The provision of more resilient back-up water supply options (e.g. groundwater) appears highly worthwhile at locations similar to location A, as it would significant improve the risk profile for these locations.

With lahars shown to be such an important component of the risk profile for farms located in close proximity to the vent and in river valleys, it is an important topic for further hazard research in Taranaki. The way in which lahar realisations were allocated to our scenarios has been based on rather broad assumptions and probability matrices, albeit based on some prior research of the hazard. With more developed hazard information we could expect some more nuanced findings for the different farms and locations, although this might also require some modifications to our modelling approach. One aspect in which the approach could be more nuanced is in relation to the timing of lahars, with these potentially being treated as separate events. It might also be possible to define different recovery times depending on the material deposited from, and potentially the velocity, of a lahar. Another aspect worth considering is the community or governance decisions that might be put in place in response to lahar risk, such as evacuations, and the resulting economic implications for farms.

## 6.2 Location versus farm

To help in understanding the importance of location relative to farm characteristics in determining economic risk, the Box and Whisker plots in Figure 9 describe the distributions of economic impacts across events, with the model run for all farms at all locations. Here the economic impact of an event is derived as the (non-discounted) difference between the sum of the Annual

---

[8]Single event scenarios where the recovery phase extends beyond the 50-year simulation have also been excluded.





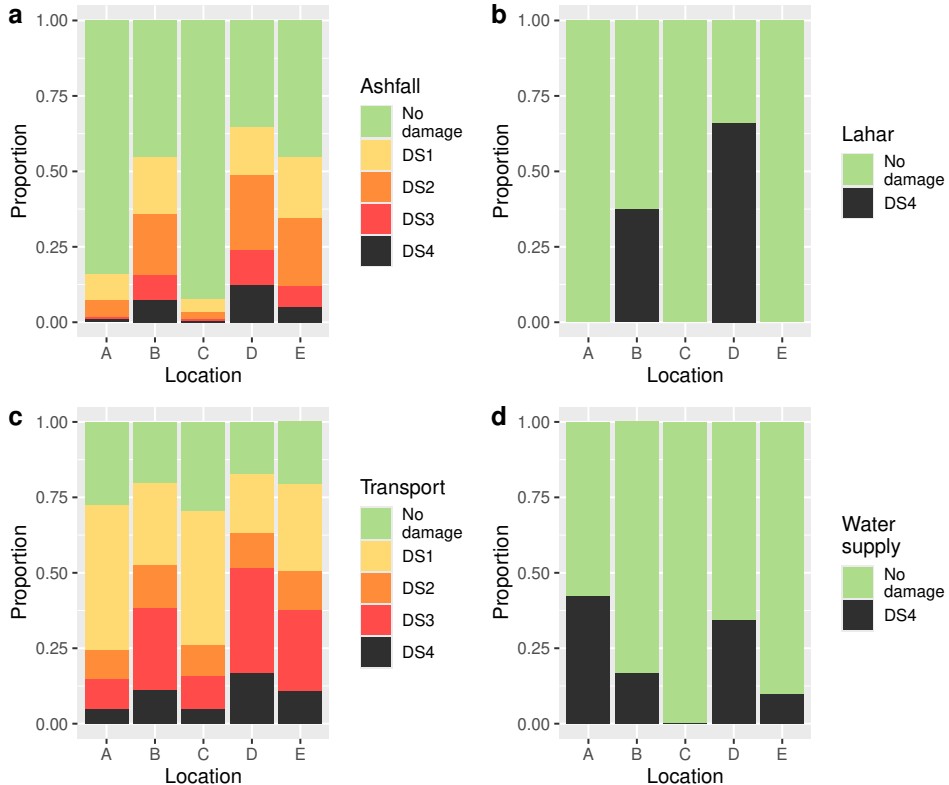

**Figure 8.** The proportion of damage states among single event simulations for locations A-E: a) ashfall damage states, b) lahar damage states, c) transport damage states, and d) water supply damage states

Net Revenues ($\sum_i \mathbf{REC}_i$), as calculated for the scenario in which the subject event occurs, and the scenario where there is no volcanic event. To help adjust for the influence of farm size on the resulting economic impacts, the economic impacts are
reported per hectare.

It is apparent from the figure that variation among farms has a small influence on economic impacts compared to farm location. For locations A, C, and E there is the classic "long tail" distribution of event impacts. This distribution shape with a high proportion of zero or close to zero losses occurred despite all scenarios having volcanic events. In particular, for Locations A and C, all farms exhibit a median event impact of zero loss. At location E, the median event impact was slightly higher,
reflecting the location's greater ashfall and transportation risks, but still less than NZ\$2,000·ha$^{-1}$ of loss across all farms. Each of these three sites nevertheless has the potential to experience substantially higher economic impacts with the rare but significant events represented by the outliers in Figure 9 of over NZ\$10,000·ha$^{-1}$.

At Locations B and D, the incidence of lahars causes distributions with a significantly different profile. At Location D, with lahars occurring across more than half of the events, and the calculated economic impacts of such events all being the same,
we see both the median and 75th percentile economic impact falling at the ends of the respective distributions for each farm.





Location B also has a large proportion of scenarios with severe impacts, and indeed the 75th percentile economic impacts are the same as calculated for Location D (ie. ranging from \$28,000 to \$38,000·ha$^{-1}$ across the 5 farms).

While there is generally little variation among the farms when placed at the same location, we can note that at location A the 75th percentile economic impact for Farms 2, 4 and 5 is less severe than that of the other farms. Given that we have already
controlled for farm size, these lower impacts partly reflect the lower stocking rates on these farms, meaning there are lower costs for asset and stock replacements. Farm 4 is also shown to experience lesser impacts at other locations, with for example the 75th percentile economic impact for Farm 4 the least severe at locations B and D. To a large extent this reflects that Farm 4, being the largest farm, has the most moderate revenue per hectare (i.e. the lowest per ha EBITDA) and so there is simply less to loose per hectares when the farm is out of operation, as well as low costs per ha to recover assets and replace stock.

In this analysis we have accounted for farm heterogeneity in terms of base characteristics of the farm before an event, e.g stock numbers, proportion of feed imported, cost and revenue structures. In terms of recovery pathways following an event, we have simply incorporated the best approximation of an "average recovery curve" for farms and assumed a farm always returns to its prior state. However, given the observation that management decisions have a significant influence on farm performance under business-as-usual conditions, it is reasonable to propose that management decisions might also have an influence on
economic performance during recovery. Thus, further research could be devoted to identifying the range of potential responses to an event and quantifying the economic implications. Although there might be little ability for farmers to avoid damages to farm natural and physical capital, there are potentially choices around the management of stock (e.g. percentage of stored feed, feed shelters), investment in land rehabilitation and selection of alternative types of management practices following an event to accelerate recovery, including even selection of alternative farm types.





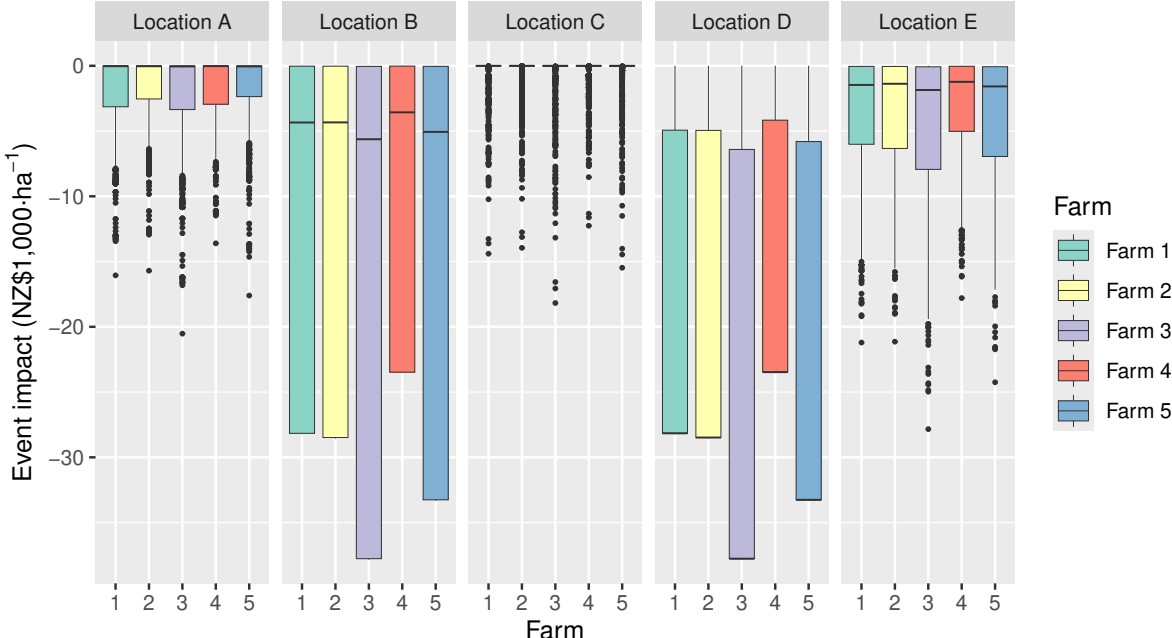

**Figure 9.** Economic impact of of single volcanic events as measured by total change in Net Annual Revenues, across five farms (1-5), at five locations (A-E).

### 6.2.1 Economic impacts of volcanic hazards over the next 50 years

Figure 10 presents the distribution of economic impacts across the scenarios for the entire 50-year simulation period. Here the economic impact of volcanic hazards for a scenario is calculated as the difference between the net present value (**NPV**) of that scenario and the **NPV** of a scenario with no volcanic events. Unlike the previous Figure 9, where all five farm types were shown at each of the five locations, Figure 10 displays each farm only in its intended location. Note further that all 10,000 simulations are included in this figure, encompassing multi-event scenarios as well as the 57% of scenarios that do not involve any eruptions. Given the latter observation, the 25th percentile and 50th percentile of losses fall at zero for all farms. A further preliminary observation is that discounting does itself help to spread the impacts under the scenarios, with higher losses calculated for scenarios having events that occur in the near future compared to scenarios where events occur in the distant future.

Notably in Figure 10, farms located in the two lahar-affected areas show a longer tail of severe impacts across the simulated volcanic events. These two farms also have higher mean NPV losses, with Farm 4 at location D a mean loss of -NZ\$6,080·ha$^{-1}$, and Farm 2 at location B a mean loss of -NZ\$3,570·ha$^{-1}$. In contrast, Farm 3 at location C, which is not in a lahar-affected area, shows a lower mean NPV loss of -NZ\$540·ha$^{-1}$. Despite not being lahar-affected, Farm 5 at Location E experiences a mean impact due to its high ashfall exposure of -NZ\$1,370·ha$^{-1}$.





When it is considered that the average value of dairy cattle farms in Taranki is NZ$44,000·ha$^{-1}$ (Colliers International, 2023; Real Estate Institute of New Zealand, 2024), the risk of economic losses to dairy farms in Taranki is shown to be far from negligible. The findings underscore the varying degrees of exposure across different farm locations and the significant role of volcanic event proximity and local terrain in determining the extent of economic risks.

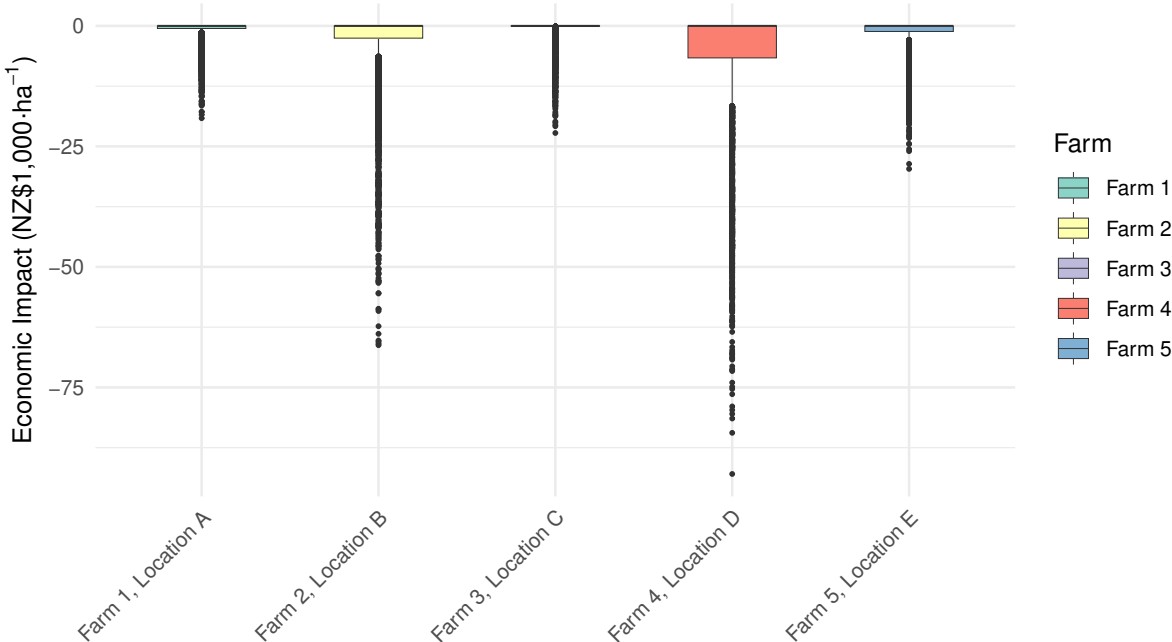

**Figure 10.** Net present value of the economic impacts of volcanic events over the next 50 years for five case study farms, as simulated under 10,000 scenarios and with a discount rate of 6% per annum.

## 6.3    Value at Risk

The Value at Risk (VaR) statistic is a useful risk management tool to assist in conceptualising risk. Essentially, it defines the worst potential losses that can be expected, over a defined time period, for a given confidence level. The ridgeline plots of probability densities in Figure 11 illustrate the placement of the VaR for a 95 percent confidence level. To enhance the visibility of the larger NPV losses, the y-axis has been transformed using a fourth root transformation. This magnifies the lower end of the distributions, making less common high-impact scenario densities more visible. The VaR for alternative confidence levels

can also be calculated (Figure 12) which is informative for decision makers and communities who will have a range of risk preferences.

    Using the 95 percent confidence level VaR, Farm 2 and Farm 4 exhibit the highest value at risk from volcanic events, primarily due to lahar impacts. Among Farms 1, 3, and 5, Farm 3 has the lowest value at risk. The value at risk across the five farms ranges from NZ$5,200·ha$^{-1}$ to NZ$35,700·ha$^{-1}$, with non-lahar affected farms reaching NZ$8,100·ha$^{-1}$, and lahar-

affected farms ranging from NZ$21,300·ha$^{-1}$ up to NZ$35,700·ha$^{-1}$.



The ridgeline plot (Figure 11) highlights that the longest distribution tails correspond to the most impacted locations. Notably, the simulation with the maximum financial impact reached NZ\$93,000·ha$^{-1}$, for the most heavily impacted Farm 4. For non-lahar affected farms, the maximum was NZ\$29,700·ha$^{-1}$ at Farm 5. The extreme cases will correspond to scenarios with not only high impact and near term events, but also the incidence of multiple volcanic events across the 50-year horizon.

**Figure 11.** Probability density functions of the economic losses from volcanic events over the next 50 years for five case study farms. Notes (1) VaR = value at risk for the 95th confidence level (2) NPV loss is the Net Present Value of total losses in Net Annual Revenues applying a discount rate of 6% per annum

In Figure 12 we plot the VaR for increasing confidence levels. Due to the predominance of no-eruption scenarios, the VaR for all five farms remains zero until nearly 60 percent confidence. After this point, it begins to rise, particularly for Farm 4 but




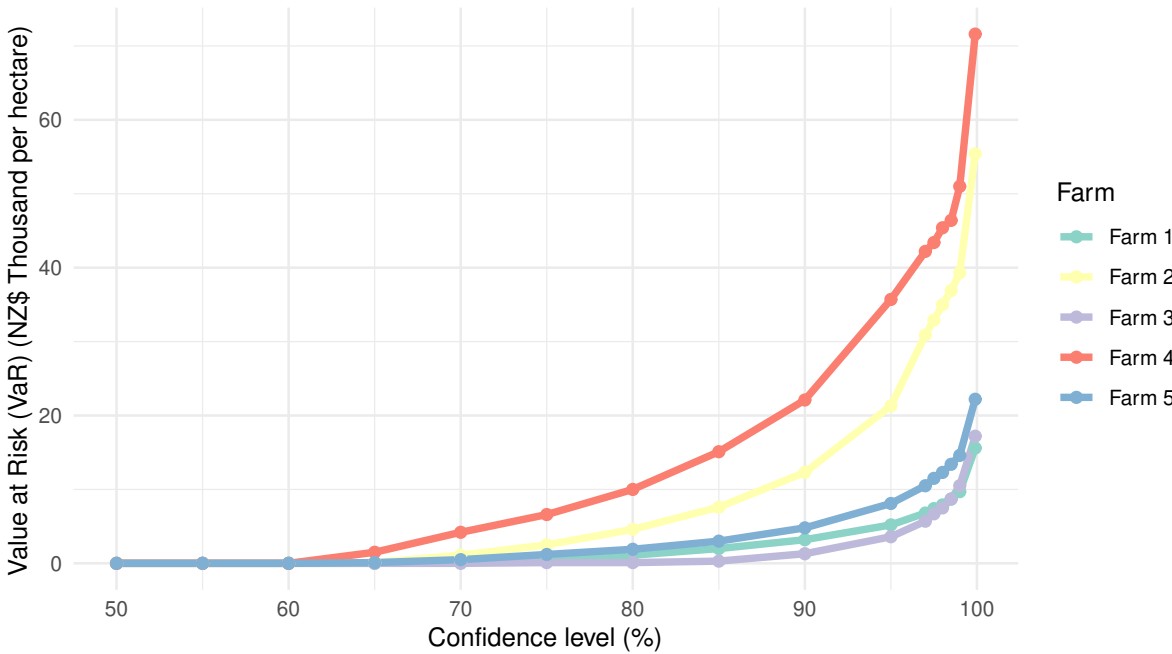

**Figure 12.** Value at risk curves for five case study farms derived from simulated NPV losses over the next 50 years

also for Farm 2. As in Figure 12, Farm 4, the most impacted farm, has a 90 percent confidence level VaR of NZ\$22,100·ha$^{-1}$, increasing to NZ\$71,600·ha$^{-1}$ at 99.9 percent confidence. For the non-lahar affected Farm 5, the 90 percent confidence VaR is NZ\$4,800·ha$^{-1}$, increasing to NZ\$22,200·ha$^{-1}$ at 99.9 percent confidence. The results for Farm 3 indicate that farms located
away from the main ashfall areas to the east of the volcano and with low lahar risk experience losses only in the worst 10-15 percent of scenarios (VaR confidence over 85 percent).

## 7 Conclusion

The quantification of economic risks to farms or other businesses from volcanic hazards is a multi-disciplinary, system-oriented exercise. We are required to pass appropriate information and research findings from the underlying physical volcanic pro-
cesses, through models of hazard generation and models capturing cascading failures of support systems, prior to being able to even consider the dynamics of business economic impacts and recovery. This study has nevertheless provided a successful demonstration of such an "end-to-end" endeavour. It has also produced a agricultural business model that is sufficiently general to be applied in other similar contexts.

There are opportunities to refine the analysis, for example through integration of more developed lahar hazard information
or more nuanced consideration of recovery costs for overlapping events. Regardless, the findings presented in this paper are





certainly among the most informative that are available to assist farmers and other decision makers in understanding the risks to dairy cattle farming from volcanic hazards in the Taranaki Region.

We have shown that dairy cattle farms in Taranaki region are generally sufficiently similar such that variations in farm management will likely have a small influence on economic risk when compared to the variation in risk attributed simply to
spatial variation in hazard exposure. For the three case study farms that had negligible lahar exposure, the modelling results indicate that, with a level of 90% confidence, the economic impacts from volcanic events over the next 50 years will not exceed approximately 10% of the property value of those farms. By comparison, in the case of the farm with the most significant lahar and ashfall exposure, the results indicate that we cannot be more than around 80% confident that the impacts over the next 50 years will not exceed around one-quarter of the farm's value, and we also cannot be more than around 90% confident that the
impacts will not exceed approximately half of the farm's value. These results indicate that, provided the region's dairy sector has sufficient access to risk information, we should anticipate volcanic risk as having a significant influence on investment choices and other management decisions that shape the future spatial evolution of the sector.

In terms of future research, the modelling pipeline and risk metrics demonstrated in this paper could be used to assess mitigation and adaptation strategies to reduce the risk from volcanic hazards, as well as to improve the preparedness and resilience
of farms. For example, we could consider the appropriateness of *ex ante* self-insurance strategies such as diversification of income sources, increasing the share of capital dedicated to off-farm activities, or geographically spreading the risk by investing in farms outside of the risky zones (Choumert-Nkolo et al., 2021). In these regards the value-at-risk metric used in this study for a "self-insurance scenario(s)" would be estimated and compared against the "no mitigation scenario" as part of an evaluation of the merits of mitigation strategies.


*Data availability.* Impact Scalars ($S_j^0$) are available in a supplementary file.

*Author contributions.* N McDonald designed the content of the study and organised the input and data transfers from other persons, with some assistance from G McDonald, T. Wilson and A. Weir. N McDonald also designed the AgriBBM model and developed associated code,
while E. Harvey and L. Dowling assisted with code development and refinement, and L. Dowling ran simulations. N McDonald, L Dowling, H Craig and D Walker participated in information gathering and parameter setting. A Weir and M Bebbington developed the method for generating volcanic scenarios and N Bui and C Magill assisted in generating the scenarios. N Bui also undertook the necessary road network modelling. N McDonald prepared the manuscript with contributions from E Harvey, L Dowling, S Cronin and J Monge. L Dowling and



N Bui compiled figures. All authors contributed to review of the manuscript. G McDonald and S Cronin were principally responsible for
winning the grant that enabled this research to take place.

*Competing interests.* The authors declare that they have no conflict of interest.

*Acknowledgements.* We would like to acknowledge the funding support from the New Zealand Government's Ministry of Business, Innovation and Employment (MBIE) under the "He Mounga Puia: Puea Rū, Puea Kōrero" or "Transitioning Taranaki to a Volcanic Future"
Endeavour Research Programme (contract UOAX1913). We also acknowledge the valuable assistance provided by M Newman in selecting appropriate representative farms at each case study location.



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





## Appendix A: Fragility functions

### A1 Ashfall

Ashfall damage states are described as:

**DS0** No disruption

**DS1** Some disruption

**DS2** Moderate disruption

**DS3** High disruption

**DS4** Total loss of capabilities

We use the piecewise linear fragility function parameters for Large Pastoral Farms from Craig et al. (2021), with the additional assumption that for ashfall depths over 1000mm we assume the damage state is always DS4. These are shown in Figure A1a and the values for the points on the graph given in Table A1.

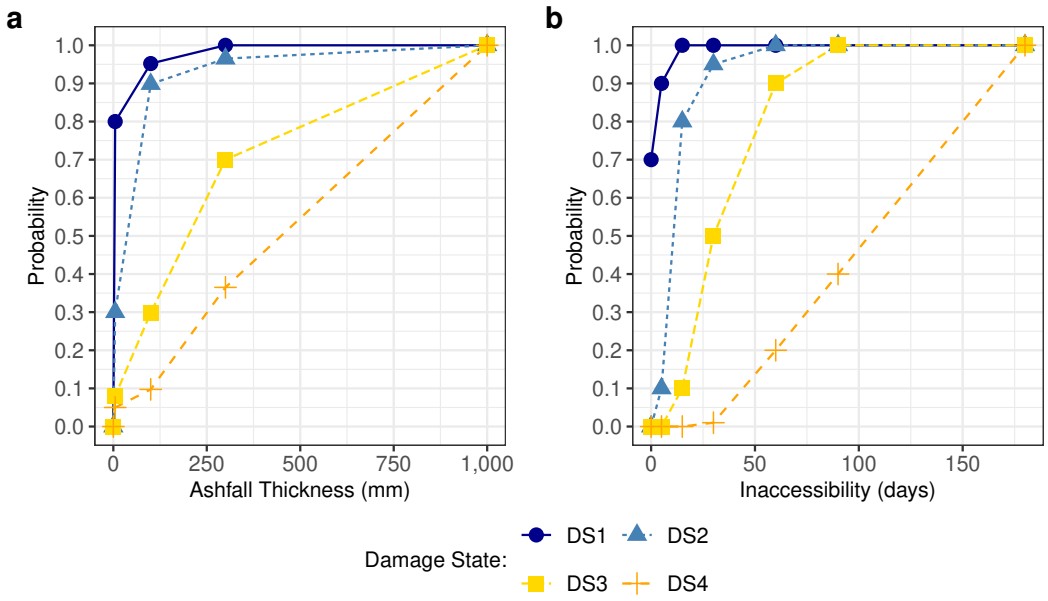

**Figure A1.** Probability of damage states due to a) ashfall thickness and b) days of road inaccessibility. Note here that even for a situation with no time inaccessible (inaccessibility = 0 days), there is still a high probability of being in DS1, due to wider network disruptions.

### A2 Transport

The five Transport Damage States are described as:





**Table A1.** Boundary values for the piecewise smoooth Ashfall fragility functions.

|  | Ashfall thickness (mm) | | | | |
|---|---|---|---|---|---|
|  | 0 | 3 | 100 | 300 | 1000 |
| Pr(DS≥DS1) | 0 | 0.8 | 0.95 | 0.97 | 1 |
| Pr(DS≥DS2) | 0 | 0.3 | 0.9 | 0.95 | 1 |
| Pr(DS≥DS3) | 0 | 0.07 | 0.3 | 0.7 | 1 |
| Pr($DS$≥DS4) | 0 | 0.04 | 0.1 | 0.35 | 1 |

**DS0**    No impact

**DS1**    Some disruption - additional costs faced by farms

**DS2**    Moderate disruption - some output from farms cannot get to market

**DS3**    High disruption - feed supplies depleted, some stock culled, loss of output to market, cows dried off

**DS4**    Severe disruption - farms cease operations, cows culled

We have developed piecewise linear fragility functions for Dairy farms that depend on the number of days that the farm is completely inaccessible by road due to ashfall impacts on the roads. These fragility functions are shown in Figure A1b, and the values in the plots are given in Table A2.1. It is important to note here that even for a situation with no time inaccessible (inaccessibility = 0 days), we assume that there is still a high probability of being in DS1, due to wider network disruptions. This is different to many other fragility functions in the literature.

After calculating the initial probability of being in each damage state from Figure A1b, the probability for each damage state is then adjusted downwards for transport disruptions that extend over the lower vulnerability period. See Table **??** for details.

**Table A2.1.** Boundary values for the piecewise smooth Ashfall fragility functions, and timing and adjustment parameters for Low Vulnerability Period(s)

|  | Transport inaccessibility (days) | | | | | | |
|---|---|---|---|---|---|---|---|
|  | 0 | 5 | 15 | 30 | 60 | 90 | 180 |
| Pr(DS≥DS1) | 0.7 | 0.9 | 1 | 1 | 1 | 1 | 1 |
| Pr(DS≥DS2) | 0 | 0.1 | 0.8 | 0.95 | 1 | 1 | 1 |
| Pr(DS≥DS3) | 0 | 0 | 0.1 | 0.5 | 0.9 | 1 | 1 |
| Pr(DS≥DS4) | 0 | 0 | 0 | 0.01 | 0.2 | 0.4 | 1 |
| Low Vulnerability Adjustment for disruptions outside milking period | | | | | | | |
| 28 May - 2 August | 50% reduction* | | | | | | |

Notes: * This is the maximum adjustment. The adjustment becomes less when only a portion of the inaccessibility phase occurs in the lower vulnerability period.





In addition to the disruption caused by ashfall on the roads, in this region, we also need to take into consideration the transport

disruptions caused by bridges on key roads being impacted by lahars. Here we consider bridges on the State Highway network that cross 9 principal river channels. If any one or more of the bridges on route to any regional access point are damaged, we increase the probability of being in DS1. Then, if at least one bridge on each side of the farm is damaged, this prevents access out of the region, and we increase the likelihood of being in DS2 and of being in DS3. The probabilities vary by farm location, and are given in Table reftab:laharTransportImpact.

**Table A2.2.** Transport Damage State probabilities due to lahar impacts on bridges on the State Highway network, for different locations

| Experienced Damage* | Damage State | Probability of being in Damage State or higher due to lahar impacts on bridges | | | | |
|---|---|---|---|---|---|---|
| | | Location A | Location B | Location C | Location D | Location E |
| Bridge on river channel on route to any regional access point | DS1 | 95% | | | | |
| Bridge on river channels on both sides of farm, preventing access out of region | DS2 | 82% | | | | |
| Bridge on river channels on both sides of farm, preventing access out of region | DS3 | 8% | 16% | 49% | 41% | 8% |

Notes: * Only bridges on State Highways crossing 9 principal river channels considered.

The final transport Damage State probabilities are the maximum of the inaccessibility due to ashfall fragility function calculation (table A1) and the lahar bridge impact (table A2.2).

## A3   Lahar

The two damage states for lahar impact are:

     **DS0**    Limited or no impact from lahar

     **DS4**    Majority of farm directly impacted from lahar.

Here we simply set the damage state to DS0 if the farm is not directly hit by a lahar, and DS4 if it is.

## A4   Water Supply

We only consider two damage states for loss of water supply:

     **DS0**    Limited or no disruption.

     **DS4**    Severe disruption - no water available for long enough that the farm has to dry off or sell cows and cease operation.

Here we applied a simple threshold function, where if there is no water supply for 30 days or more, the farm is in DS4 and

otherwise DS0.





## Appendix B: Hazard Realisation

### B1 Lahar Realisation

Given the distribution of ashfall from an event, we used the probabilities specified in Tables B1.1 and B1.2 to respectively assign the realisation of lahar to each farm and principal river valley.

**Table B1.1.** Probability that farm will be impacted by lahar

| Farm | Maximum Ash Depth at park boundary (mm) | Degrees between farm from maximum depth | | |
|---|---|---|---|---|
| | | <45 degrees | 45-90 degrees | >90 degrees |
| Farm 2 | 0-50 | 0.30 | 0.20 | 0.00 |
| | 50-200 | 0.80 | 0.60 | 0.10 |
| | 200-500 | 1.00 | 0.80 | 0.50 |
| | 500+ | 1.00 | 1.00 | 0.80 |
| Farm 3 | 0-50 | 0.00 | 0.00 | 0.00 |
| | 50-200 | 0.00 | 0.00 | 0.00 |
| | 200-500 | 0.50 | 0.05 | 0.05 |
| | 500+ | 0.90 | 0.10 | 0.05 |
| Farm 4 | 0-50 | 0.70 | 0.50 | 0.20 |
| | 50-200 | 0.90 | 0.70 | 0.30 |
| | 200-500 | 1.00 | 1.00 | 0.80 |
| | 500+ | 1.00 | 1.00 | 1.00 |

**Table B1.2.** Probability that river will be impacted by lahar at point of State Highway crossing

| River | Maximum Ash Depth at park boundary (mm) | Degrees between River from maximum depth | | |
|---|---|---|---|---|
| | | <45 degrees | 45-90 degrees | >90 degrees |
| Stony | 0-50 | 0.40 | 0.10 | 0.00 |
| | 50-200 | 0.80 | 0.70 | 0.20 |
| | 200-500 | 1.00 | 1.00 | 0.80 |
| | 500+ | 1.00 | 1.00 | 1.00 |
| Other | 0-50 | 0.10 | 0.05 | 0.00 |
| | 50-200 | 0.30 | 0.10 | 0.00 |
| | 200-500 | 0.80 | 0.60 | 0.30 |
| | 500+ | 1.00 | 1.00 | 0.80 |



## B2 Water Supply Failure

The length of time for water supply failure was determined from ashfall thickness at the water supply nodes relevant to the water supply type; municipal, surface water collection, or groundwater collection as shown in Figure B1. We assigned supply types to farms based on location relative to municipal supply schemes and suitable surface water sources, following Wild (2016). Damage functions were derived from information on municipal supply impacts in Wild (2016) and from farm impacts described in Porter (2022), Wilson et al. (2014), Thompson et al. (2017), and Neild et al. (1998). Specifically, the function for estimating days without water supply for municipal systems depended on the ashfall thickness at the farm, municipal supply intake, and local power transformer station. Meanwhile, the functions for estimating days without surface and groundwater supply were based on the ashfall thickness at the farm.

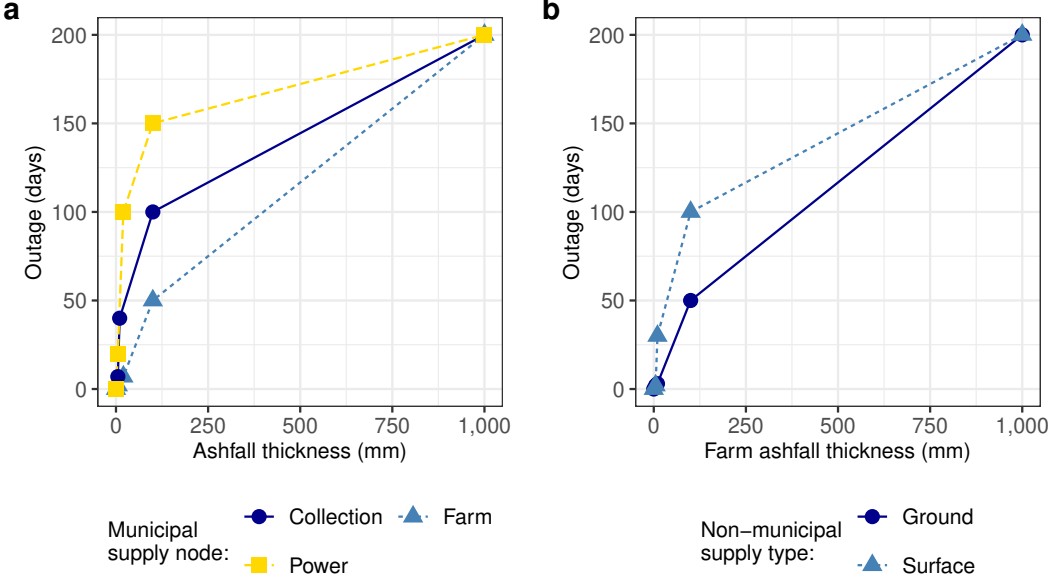

**Figure B1.** Water availability outage length in days as a function of ashfall thickness for a) municipal water supply as the maximum delay across delays at the water collection point, farm, and local power source (power station), and b) ground and surface water sources dependant on ashfall thickness at the farm.

## Appendix C: Impact States and Impact Duration

The mapping between damage states, impact states and methods used to determine impact duration is provided in Table C1. For each impact state, the share of stock remaining on site is also provided in Table C1. Note these shares were important in deriving the impact scalars for each impact state. Due to the size of the matrices, the full set of impact scalars is provided in the Supplementary Material. The function that was used to approximate impact duration given ashfall depth (DF), is described in



Figure C1. Impact duration was based on descriptions and data on recovery from previous eruptions (Wilson et al., 2011a, b;
U.S. Geological Survey, 2024; Dale and Crisafulli, 2018; del Moral and Wood, 1993).

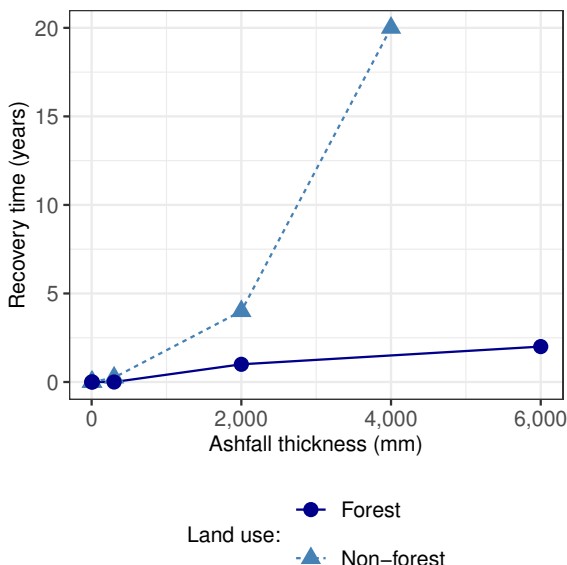

**Figure C1.** Event impact duration as a function of ashfall thickness for forest and non-forest land uses.





**Table C1.** Mapping Between Damage States, Impact States and Impact Duration

| Damage State | | | | Impact State | Impact Duration | Share of stock remaining |
|---|---|---|---|---|---|---|
| Lahars | Transport | Water Supply | Ash | | | |
| DS0 | DS0 | DS0 | DS0 | IS0 | 0 yr | 1.00 |
| DS0 | DS0-DS1 | DS0 | DS1 | IS1A | 1 yr | 1.00 |
| DS0 | DS0,DS1 | DS0 | DS2 | IS2A | DF | 0.80 |
| DS0 | DS0,DS1 | DS0 | DS3 | IS3A | DF | 0.40 |
| DS0 | DS0-DS4 | DS0 | DS4 | IS4 | DF | 0.00 |
| DS0 | DS0-DS4 | DS4 | DS0-DS2 | IS4 | W | 0.00 |
| DS0 | DS0-DS4 | DS4 | DS3-DS4 | IS4 | WDF | 0.00 |
| DS0 | DS1 | DS0 | DS0 | IS1T | 0.5 yr | 1.00 |
| DS0 | DS2 | DS0 | DS0 | IS2T | IP | 1.00 |
| DS0 | DS2 | DS0 | DS1 | IS1AIS2T | 1 yr | 1.00 |
| DS0 | DS2 | DS0 | DS2 | IS2AIS2T | DF | 0.76 |
| DS0 | DS2 | DS0 | DS3 | IS3AIS2T | DF | 0.30 |
| DS0 | DS3 | DS0 | DS0,DS1 | IS3T | 1 yr | 1 - $ifp$ |
| DS0 | DS3 | DS0 | DS2 | IS2AIS3T | 1 yr | 0.70 |
| DS0 | DS3 | DS0 | DS3 | IS3AIS3T | DF | 0.30 |
| DS0 | DS4 | DS0 | DS0 | IS4 | 1 yr | 0.00 |
| DS0 | DS4 | DS0 | DS1,DS2 | IS4 | IP | 0.00 |
| DS0 | DS4 | DS0 | DS3,DS4 | IS4 | DF | 0.00 |
| DS4 | DS0-DS4 | DS0,DS4 | DS0-DS4 | IS4 | LDF | 0.00 |

Notes: 0 yr = 0 years, 0.5 yr = 0.5 years, 1 yr = 1 year, DF = Ash Depth Function, IP = Inaccessibility Period, W = Water Supply Re-establishment, WDF = Maximum of Ash and Water, LDF = Maximum of Lahar and Ash, $ifp$ = proportion of farm feed that is normally imported

## Appendix D: Recovery/clean up costs

**Asset replacement costs**:

The damage functions for farm assets are described in Figure D1 while the parameters establishing the replacement or rehabilitation costs for assets are provided in Table D1





**Table D1.** Recovery Costs for Farm Assets

| Asset Type | Replacement or Re-habilitate | Value | Unit | Comments |
|---|---|---|---|---|
| Milking shed | Replacement | 710,000 - 920,000 | $NZ per farm | Based on $500,000 + $10,000 for every 10 cows |
| Other sheds | Replacement | 300,000 | $NZ per farm | Value of sheds and their contents |
| Farm dwelling | Replacement | 700,000 | $NZ per farm | Based on replacement costs for a 3 bed-room home plus study in Taranaki (Vero Insurance New Zealand, 2024) |
| Machinery | Replacement | 80,000 | $NZ per farm | Cost of tractor and implements |
| Fences | Replacement | 1,600 | $NZ per ha | Assumes 200 meters of fencing per hectare at a rate of $8 per meter (Ministry for Primary Industries, 2016) |
| Land | Rehabilitate | 5,600 | $NZ per ha | Based on average per ha earthworks costs for contouring land to establish an orchard, plus 3 times usual annual grass fertilizer application |

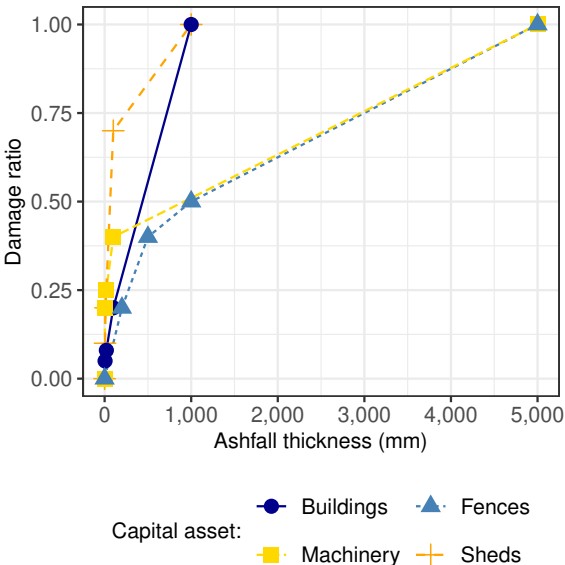

**Figure D1.** Damage ratios used for calculating losses as a proportion of asset value. Damage ratios were a function of ashfall thickness (mm) at the farm determining losses for buildings, fences, machinery, and sheds.