# Peer review of "Quantifying economic risks to dairy farms from volcanic hazards in Taranaki, New Zealand"

_EGUsphere, 2024_

## Author Response (AR1)

Dr. Giovanni Macedonio
Editor
Natural Hazards and Earth System Sciences

Dear Dr Macedonio

Thank you very much for your time in reviewing our paper and for accepting it for publication subject to minor revisions.

Below I have provided a summary of the changes that have been made to the revised manuscript. I have also noted down the original Referee comments below so that it is easy to identify how the changes match to the comments received.

We have written the manuscript in Overleaf and used tracked changes. The Copernicus system has only allowed us to upload a Pdf for the mark up version. The significant changes are therefore all identified in the mark-up version of the Pdf in red.

**Referee 1**

*[Comment 1] Agricultural activities are seasonal and the impact of an event (and the likely management decisions made) will likely depend on the time of year that the hazard occurs. This does not seem to be discussed in the paper. This needs to be discussed as a limitation or opportunity for future research at the very least.*

As explained in the original response to Reviewer 1, we agree that seasonality is an important matter to consider for agriculture. Some aspects of seasonality were considered in the modelling. We have made these considerations clearer to the reader, and added additional text to highlight the importance of seasonality:

- Revised wording in the bullet points around line 317 (originally line 310) to explicitly call this a seasonal consideration.
- Sentence added in Appendix A2 – see line 758.
- Original footnote 6 added to the main text – see around line 301.
- A whole paragraph has been added to the conclusion section – see line 585.

*[Comment 2] It appears that farm impacts (and subsequent recovery) are based on 'actual' hazard impacts and are reactive. Were pre-emptive adaptations e.g. relocation of stock ever considered? Perhaps add to discussion around line 500 (so to include not just reactive but pro-active actions when warnings are issued).*

As noted in our original response, one point to note is that the assignment of ashfall damage states is based on the study by Craig et al (2021). To the extent that the farms surveyed in that study undertook pre-emptive actions, we will have given some account to pre-emptive measures in the calculations of the impacts.

In our opinion the two topics most worthy of consideration for improving the impact calculation in relation to pre-emptive measures relate to actions taken to preserve human life and health, ie.

- Evacuations during and prior to an eruption when volcanic unrest is high
- Evacuations when lahar risk is considered high

We have edited the manuscript to be clearer in the text is worthy of future research:

- Some edits to the paragraph around line 512 (originally line 500) to highlight the idea of pre-emptive measures as potentially being important, including providing the example of pre-emptive stock evacuations as an example of a pre-emptive management strategy.
- The last conclusions paragraph has been edited to explicitly mention pre-emptive land use change in high risk areas as an area of future research.
- The last paragraph in the conclusion now also draws attention to the importance of life-safety considerations and the need to consider farm disruptions caused by evacuations in anticipation of an event.

*[Comment 3]* It was not clear how much, if at all, post-event commodity price changes are considered (both impact and recovery costs)? Increased transportation costs were discussed but not commodity prices themselves. It might be helpful to comment on this.

We noted in our original response that price changes were considered regarding stock sales. In addition, we have edited the manuscript:
- The paragraph around line 104 has been expanded to ensure readers understand that the impact scalars can capture both price and quantity impacts.

*[Technical Corrections]*

We have corrected all of the minor errors noted in Reviewer 1's comments:

- The caption for Figure 1 has been revised so that it is clearer and subscripts $i$ and $j$ are defined.
- Additional text added around line 90 so that text better relates to Figure 1.
- Minor gramma and typos corrected.

**Referee 2**

*[Comment 1] The consideration of dangers for a dairy operation were detailed however no consideration for potential injury or death of workers or operators seems to have been considered. In particular, would farms simulated to be affected directly by lahars (or pycroclastic flows as proxied by lahars in the model) have any associated risk to humans. Similarly, would ash fall have any consequences for human health? Injury and/or loss of life would be tragic, costly and detrimental for returning a farm back to full operating capacity. The conclusion of the report describes the findings as informative for farmers and decision-makers in understanding the risks to dairy farming. I would consider any potential impact on human health or life to be key information for farmers in this area of inquiry.*

*Some explanation of why these elements were not considered, why the volcanic impacts are not considered to pose a risk to farm workers or operators, or acknowledging any limitations in the method for considering these elements would strengthen the paper.*

As noted in our original response, we agree that risks to human life and health, including for farmers, are important considerations in the context of a future Taranaki eruption. In terms of economic impacts, we believe that it will be actions taken to reduce these risks (e.g. farm evacuations) that will be particularly important to consider. Please see notes for Comment 2 of Referee 1.

*[Comment 2] Similarly, in the cases of 100% destocking with the event of lahars, is the assumption that this is due to animal mortality? If so, does the model consider no recovery value for those animals (where in other cases some recovery can be achieved by selling stock)? Clarifying this would be appreciated.*

We have added more information into Table C1in the appendix to clarify the parameters used in the modelling which determine recovery value for stock. A sentence has also been added in the text of Appendix C

*[Comment 3] In the case of recovery prices for the replacement of farm assets, do these costs include the demolition and/or clearing costs for removing the assets they are replacing. Clarification or acknowledgement of any limitations in this area would be appreciated.*

We have reviewed the data source from which we obtained some of the farm asset replacement costs and some demolition costs were included. We have noted this at line 423.

*[Comment 4] The explanation of why impact duration (or recovery time) was used as the proxy for severity in the event of overlapping events could be expanded. Are there cases of longer term but minor impacts or conversely short term but high impacts? Would it be possible to instead, use average impact over the remaining years as a proxy of severity? Slightly more explanation or acknowledgement of the minor limitation would be appreciated.*

A footnote has been added at line 205.

*[Comment 5] I had difficulty in understanding how I was meant to interpret Figure A1b in terms of the relationship between the two axes and which measure was determining what. Some additional clarification on how to read this graph would be appreciated.*

The first part of the caption of Figure A1 has been adjusted so that the Figure can now be interpreted.

*[Comment 6] No source was provided for the data in Figure 4e.*

We have now included a reference for the source data.

*[Comment 7] Why are the total recovery costs spread evenly over the impact duration. Would traditional farm assets (i.e. dairy platform, shed, house) not have a standard loan and repayment schedule or are these considered in the impact duration?*

We have added text in a footnote around line 130 to make it clear that this is a simplifying assumption.

*[Technical Corrections]*

The much greater than notation was intentional.

Other errors noted by Referee 2 have been corrected in the manuscript.

As far as we are aware, we have now addressed all comments received on the paper.

Thank you again for your time and consideration of our manuscript.

Yours sincerely

Nicola McDonald